# A Unifying Framework for Online Optimization with Long-Term Constraints

**Matteo Castiglioni**[*]
Politecnico di Milano

**Andrea Celli**[†]
Bocconi University

**Alberto Marchesi**[*]
Politecnico di Milano

**Giulia Romano**[*]
Politecnico di Milano

**Nicola Gatti**[*]
Politecnico di Milano

## Abstract

We study online learning problems in which a decision maker has to take a sequence of decisions subject to $m$ *long-term constraints*. The goal of the decision maker is to maximize their total reward, while at the same time achieving small cumulative constraints violation across the $T$ rounds. We present the first *best-of-both-world* type algorithm for this general class of problems, with no-regret guarantees both in the case in which rewards and constraints are selected according to an unknown stochastic model, and in the case in which they are selected at each round by an adversary. Our algorithm is the first to provide guarantees in the adversarial setting with respect to the optimal fixed strategy that satisfies the long-term constraints. In particular, it guarantees a $\rho/(1 + \rho)$ fraction of the optimal reward and sublinear regret, where $\rho$ is a feasibility parameter related to the existence of strictly feasible solutions. Our framework employs traditional regret minimizers as black-box components. Therefore, by instantiating it with an appropriate choice of regret minimizers it can handle the *full-feedback* as well as the *bandit-feedback* setting. Moreover, it allows the decision maker to seamlessly handle scenarios with non-convex rewards and constraints. We show how our framework can be applied in the context of budget-management mechanisms for repeated auctions in order to guarantee long-term constraints that are not *packing* (*e.g.*, ROI constraints).

## 1 Introduction

We study online learning problems where a decision maker takes decisions over $T$ rounds. At each round $t$, the decision $\boldsymbol{x}_t \in \mathcal{X}$ is chosen before observing a reward function $f_t$ together with a set of $m$ *time-varying* constraint functions $g_t$. The decision maker is allowed to make decisions that are *not* feasible, provided that the overall sequence of decisions obeys the *long-term constraints* $\sum_{t=1}^{T} g_t(\boldsymbol{x}_t) \leq \boldsymbol{0}$, up to a small cumulative violation across the $T$ rounds. The problem becomes that of finding a sequence of decisions $\boldsymbol{x}_t$ which guarantees a reward close to that of the best fixed decision in hindsight while satisfying long-term constraints. This type of framework was first proposed by Mannor et al. [36], and it has numerous applications ranging from wireless communication [36] and multi-objective online classification [13], to *safe* online learning [4].

Mannor et al. [36] show that guaranteeing sublinear regret and sublinear cumulative constraints violation is impossible even when $f_t$ and $g_t$ are simple linear functions. Therefore, previous works either focus on the case in which constraints are generated i.i.d. according to some unknown stochastic

---

[*]DEIB, Politecnico di Milano, {matteo.castiglioni, alberto.marchesi, giulia.romano, nicola.gatti}@polimi.it.
[†] Computing Sciences Department, Bocconi University, andrea.celli2@unibocconi.it.

36th Conference on Neural Information Processing Systems (NeurIPS 2022).

| Algorithm | Constr. | Non-convex $f_t$ and $g_t$ | Bound — constant $\rho$ | | Bound — arbitrary $\rho$ | |
|---|---|---|---|---|---|---|
| | | | Reward | Violation | Reward | Violation |
| Yu et al. [44] | STOC | ✗ | $\mathrm{OPT} - \tilde{O}(T^{1/2})$ | $\tilde{O}(T^{1/2})$ | — | — |
| **Ours** | STOC | ✓ | $\mathrm{OPT} - \tilde{O}(T^{1/2})$ | $\tilde{O}(T^{1/2})$ | $\mathrm{OPT} - \tilde{O}(T^{3/4})$ | $\tilde{O}(T^{3/4})$ |
| | ADV | ✓ | $\frac{\rho}{1+\rho}\mathrm{OPT} - \tilde{O}\left(T^{1/2}\right)$ | $\tilde{O}(T^{1/2})$ | — | — |

Table 1: Comparison between the performance of our algorithm and previous work using the same baseline as ours. Bounds for settings that were *not* previously tractable are highlighted in gray. OPT is the reward of the baseline.

model, without providing any guarantees for the adversarial case, or provide results for adversarially-generated constraints under some strong assumptions on the structure of the problem or using a weaker baseline (a detailed discussion of related works can be found in Appendix A). A few examples in the latter case are [16, 21, 40, 42]. In the former setting (*i.e.*, stochastic constraints), Wei et al. [41] consider a weaker baseline that is feasible for each constraint $g_t$, going against the basic idea of long-term constraints. A notable exception is the work by Yu et al. [44], who employ the same baseline as ours, and provide an upper bound of $\tilde{O}(T^{1/2})$ for both regret and constraints violation (see Table 1). We also mention that there are some works studying the problem in which constraints are *static* (see, *e.g.*, [31, 35, 43, 45]), or focus on specific types of constraints, such as *knapsack constraints* [7, 30]. Our framework differs from those works as we deal with *arbitrary* and *time-varying* constraints. Moreover, it also extends the *online convex optimization* framework introduced by Zinkevich [46] by allowing for general non-convex loss functions $f_t$, arbitrary feasibility sets $\mathcal{X}$, and arbitrary time-varying long-term constraints.

## 1.1 Original contributions

Given the negative result by Mannor et al. [36], a natural question is what kind of guarantees we can reach in the adversarial setting, when adopting the standard baseline of the best fixed decision in hindsight satisfying (in expectation) the long-term constraints. We provide the first positive result going in this direction, by designing a no-$\alpha$-regret algorithm that guarantees a sublinear cumulative constraints violation. Moreover, we make a step forward in the line of work initiated by Bubeck and Slivkins [14], by showing that our algorithm is also the first *best-of-both-worlds* algorithm for problems with arbitrary long-term constraints. This allows our algorithm to guarantee good worst-case performance (adversarial case), while being able to exploit well-behaved problem instances (stochastic case). The only assumption which we require is the existence of a decision that is strictly feasible with respect to the sequence of constraints. We denote by $\rho$ the "margin" by which this decision is strictly feasible (see Section 2 for a definition). At the same time, we show that even without this assumption, we can recover sublinear regret and violation with stochastic constraints.

Previous work usually assumes that $\rho$ is a given *constant*. In that case, our algorithm matches the guarantees by Yu et al. [44] when constraints are generated i.i.d. according to an unknown distribution, and has no-$\alpha$-regret with $\alpha = \rho/(1 + \rho)$ in the adversarial case (see Table 1). Our algorithm only requires a lower bound on the real value of the feasibility parameter $\rho$. In the stochastic case, the lower bound may even be unknown, and the algorithm can efficiently estimate it from data. Moreover, we argue that if $\rho$ is allowed to depend on $T$ and take arbitrarily small values, then there are certain values ($\rho \leq T^{-1/4}$), for which any regret bound depending on $1/\rho$ would be useless (*i.e.*, *not* sublinear in $T$, see Section 3). This setting is usually overlooked by previous work, which assumes $\rho$ to be a given constant. We show that, in the case of an arbitrary feasibility parameter $\rho$, in the stochastic setting our algorithm guarantees an upper bound of $\tilde{O}(T^{3/4})$ for regret and cumulative constraints violation.

Our framework employs traditional regret minimizers as black-box components. Therefore, by instantiating it with an appropriate choice of regret minimizers it can handle *full-feedback* as well as *bandit-feedback* settings. In the former case, after playing $\boldsymbol{x}_t$, the decision maker gets to observe $f_t$ and $g_t$, while in the latter case only the realized values $f_t(\boldsymbol{x}_t)$ and $g_t(\boldsymbol{x}_t)$ are observed. Moreover, this allows the decision maker to seamlessly handle scenarios with non-convex reward and constraints, by employing a suitable regret minimizer for non-convex losses (see, *e.g.*, [39]). Our algorithm is based on a two-stage approach in which *primal* and *dual* players interact through *Lagrangian games*.

In the first (*play*) phase, the primal player tries to balance out the maximization of their rewards with constraints violation. In the second (*recovery*) phase, the primal player only makes "safe decisions" to avoid violating constraints too much. It is possible to prove that, in the case of stochastic rewards and constraints, the algorithm never enters phase two. This property is particularly relevant for budget-pacing mechanisms in repeated auctions, since it is related to how budget is allocated. Our framework can also be instantiated to perform budget allocation subject to constraints that were previously *not* tractable by traditional mechanisms, such as ROI constraints [11, 22].

## 2 Preliminaries

The decision maker has a non-empty set of available strategies $\mathcal{X}$ (this set may be non-convex, integral, and even non-compact). In each round $t \in [T]$,[2] the decision maker first chooses $\boldsymbol{x}_t \in \mathcal{X}$, and the environment selects a reward function $f_t : \mathcal{X} \to [0,1]$ and a constraint function $g_t : \mathcal{X} \to [-1,1]^m$ conditioned on the past history of play up to time $t-1$ (*i.e.*, the environment chooses $f_t$ and $g_t$ without knowledge of $\boldsymbol{x}_t$). Notice that both $f_t$ and $g_t$ need *not* be convex. The latter specifies a set of $m$ constraints of the form $g_t(\boldsymbol{x}) \le \boldsymbol{0}$, with $g_{t,i}(\boldsymbol{x}) \le 0$ denoting the $i$-th constraint.[3] In the following, we denote as $\mathcal{F}$, respectively $\mathcal{G}$, the set of all the possible $f_t$, respectively $g_t$, functions (*e.g.*, $\mathcal{F}$ and $\mathcal{G}$ may contain all the Lipschitz-continuous functions defined over $\mathcal{X}$). At each round $t \in [T]$, the decision maker can condition their decision on prior feedbacks and on the sequence of prior decisions $\boldsymbol{x}_1, \ldots, \boldsymbol{x}_{t-1}$, but no information about future rewards and constraint functions is available.

### 2.1 Strong duality through strategy mixtures

In the following, we define the optimization problem (Problem $\mathrm{LP}_{f,g}$) which is used to define the baselines against which we compare the performances of the decision maker. Such a problem involves probabilistic mixtures of strategies in $\mathcal{X}$, which are crucial in order to recover strong duality.[4]

First, we introduce the set of probability measures on the Borel sets of $\mathcal{X}$. We refer to such a set as the set of *strategy mixtures*, denoted as $\Xi$. We endow $\mathcal{X}$ with the Lebesgue $\sigma$-algebra, and we assume that all the functions in $\mathcal{F}$ and $\mathcal{G}$ are measurable with respect to every probability measure $\boldsymbol{\xi} \in \Xi$. This ensures that the various expectations taken are well-defined, since the functions are assumed to be bounded above, and they are therefore integrable. In the following, for ease of presentation and with a slight abuse of notation, whenever we write a $\boldsymbol{\xi} \in \Xi$ in place of an $\boldsymbol{x} \in \mathcal{X}$, we mean that we are taking the expectation with respect to the probability measure $\boldsymbol{\xi}$. For instance, given $f \in \mathcal{F}$ and $g \in \mathcal{G}$, we have that $f(\boldsymbol{\xi}) = \mathbb{E}_{\boldsymbol{x} \sim \boldsymbol{\xi}} f(\boldsymbol{x})$ and $g(\boldsymbol{\xi}) = \mathbb{E}_{\boldsymbol{x} \sim \boldsymbol{\xi}} g(\boldsymbol{x})$.

Then, given two functions $f \in \mathcal{F}$ and $g \in \mathcal{G}$, we define the following optimization problem, which chooses the strategy mixture $\boldsymbol{\xi} \in \Xi$ that maximizes the expected reward encoded by $f$, while guaranteeing that the constraints encoded by $g$ are satisfied in expectation.

$$\mathrm{OPT}_{f,g} := \begin{cases} \sup\limits_{\boldsymbol{\xi} \in \Xi} & f(\boldsymbol{\xi}) \qquad \text{s.t.} \\ & g(\boldsymbol{\xi}) \le 0. \end{cases} \qquad (\mathrm{LP}_{f,g})$$

We denote by $d_g \in [-1,1]$ the largest possible value for which there exists a strategy mixture $\boldsymbol{\xi} \in \Xi$ satisfying the constraints $g(\boldsymbol{\xi}) \le 0$ by a margin of at least $d_g$. Formally,

$$d_g := \sup_{\boldsymbol{\xi} \in \Xi} \min_{i \in [m]} -g_i(\boldsymbol{\xi}). \qquad (1)$$

In order to ensure that $\mathrm{OPT}_{f,g}$ is always well defined, we assume that it is always the case that $d_g \ge 0$. Notice that, if $d_g > 0$, then Problem $\mathrm{LP}_{f,g}$ satisfies Slater's condition.

In the following, we prove some auxiliary results relating to Problem $\mathrm{LP}_{f,g}$ that will be useful in the rest of the paper. First, we introduce a Lagrangian relaxation of the problem.

---

[2]In this work, we denote by $[x]$ the set $\{1, \ldots, x\}$ of the first $x$ natural numbers.

[3]Focusing on the case $g_t(\boldsymbol{x}) \le \boldsymbol{0}$ is w.l.o.g. since any set of constraints can be represented in such a form.

[4]The optimal fixed strategy mixture provides an arguably stronger baseline than the optimal fixed strategy. In stochastic settings, this baseline is related to the best dynamic policy. In particular, if we consider the case in which the observed functions are defined as the average of functions $f_t$ and $g_t$ across the $T$ rounds, then the optimal mixture provides the same utility as the best dynamic policy (see [7] for a similar result).

**Definition 2.1** (Lagrangian Function). Given two arbitrary functions $f \in \mathcal{F}$ and $g \in \mathcal{G}$, the *Lagrangian function* $\mathcal{L}_{f,g} : \Xi \times \mathbb{R}^m_{\geq 0} \to \mathbb{R}$ of Problem $\mathrm{LP}_{f,g}$ is defined as

$$\mathcal{L}_{f,g}(\boldsymbol{\xi}, \boldsymbol{\lambda}) \coloneqq f(\boldsymbol{\xi}) - \langle \boldsymbol{\lambda}, g(\boldsymbol{\xi}) \rangle.$$

If Problem $\mathrm{LP}_{f,g}$ satisfies Slater's condition, then Theorem 1 of Chapter 8.3 in [34] readily gives us that strong duality holds even if $f$ and $g$ are arbitrary non-convex functions. Formally:

**Corollary 2.2.** *Given $f \in \mathcal{F}$ and $g \in \mathcal{G}$ such that $d_g > 0$, it holds*

$$\sup_{\boldsymbol{\xi} \in \Xi} \inf_{\boldsymbol{\lambda} \in \mathbb{R}^m_{\geq 0}} \mathcal{L}_{f,g}(\boldsymbol{\xi}, \boldsymbol{\lambda}) = \inf_{\boldsymbol{\lambda} \in \mathbb{R}^m_{\geq 0}} \sup_{\boldsymbol{\xi} \in \Xi} \mathcal{L}_{f,g}(\boldsymbol{\xi}, \boldsymbol{\lambda}) = \mathrm{OPT}_{f,g}.$$

Next, we show that, if $d_g > 0$, then strong duality holds even when we restrict the admissible dual vectors $\boldsymbol{\lambda} \in \mathbb{R}^m_{\geq 0}$ to the set $\mathcal{D}_{d_g}$, where, for any $q \in \mathbb{R}_{>0}$, we let $\mathcal{D}_q \coloneqq \{\boldsymbol{\lambda} \in \mathbb{R}^m_{\geq 0} : \|\boldsymbol{\lambda}\|_1 \leq 1/q\}$ (omitted proofs can be found in Appendix B).

**Theorem 2.3.** *Given $f \in \mathcal{F}$ and $g \in \mathcal{G}$ such that $d_g > 0$, it holds*

$$\sup_{\boldsymbol{\xi} \in \Xi} \inf_{\boldsymbol{\lambda} \in \mathcal{D}_{d_g}} \mathcal{L}_{f,g}(\boldsymbol{\xi}, \boldsymbol{\lambda}) = \inf_{\boldsymbol{\lambda} \in \mathcal{D}_{d_g}} \sup_{\boldsymbol{\xi} \in \Xi} \mathcal{L}_{f,g}(\boldsymbol{\xi}, \boldsymbol{\lambda}) = \mathrm{OPT}_{f,g}.$$

## 2.2 Stochastic vs. adversarial: baselines and feasibility

We consider several settings, differing in how functions $f_t$ and $g_t$ are selected, either *stochastically* or *adversarially*. We say that functions $f_t$ (respectively $g_t$) are selected stochastically, when they are independently drawn according to a given probability measure $\mu_\mathcal{F}$ over $\mathcal{F}$ (respectively $\mu_\mathcal{G}$ over $\mathcal{G}$). Instead, we say that functions $f_t$ (respectively $g_t$) are selected adversarially if each $f_t$ (respectively $g_t$) is chosen by an adversary based on the sequence of prior decisions, namely $\boldsymbol{x}_1, \ldots, \boldsymbol{x}_{t-1}$.

Consistently with previous work (see, *e.g.*, [36]), we compare the performance of the decision maker (in terms of reward cumulated over the $T$ rounds) against the baseline $T\,\mathrm{OPT}_{\bar{f},\bar{g}}$ (as defined by Problem $\mathrm{LP}_{\bar{f},\bar{g}}$), where $\bar{f}$ and $\bar{g}$ are suitably-defined functions. In particular:

- When functions $f_t$, respectively $g_t$, are selected stochastically, then we define function $\bar{f}$, respectively $\bar{g}$, so that $\bar{f}(\boldsymbol{x}) \coloneqq \mathbb{E}_{f \sim \mu_\mathcal{F}}[f(\boldsymbol{x})]$, respectively $\bar{g}(\boldsymbol{x}) \coloneqq \mathbb{E}_{g \sim \mu_\mathcal{G}}[g(\boldsymbol{x})]$.

- When functions $f_t$, respectively $g_t$, are selected adversarially, then we define function $\bar{f}$, respectively $\bar{g}$, so that $\bar{f}(\boldsymbol{x}) \coloneqq \frac{1}{T} \sum_{t=1}^{T} f_t(\boldsymbol{x})$, respectively $\bar{g}(\boldsymbol{x}) \coloneqq \frac{1}{T} \sum_{t=1}^{T} g_t(\boldsymbol{x})$.

Intuitively, in the stochastic case, the baseline is instantiated with an expectation of functions taken with respect to the probability measure $\mu_\mathcal{F}$ (respectively $\mu_\mathcal{G}$). Instead, in the adversarial case, the baseline uses the average of functions $f_t$ (respectively $g_t$) observed over the $T$ rounds.

Let us remark that, when the set $\mathcal{X}$ is compact convex and functions $f_t$ and $g_t$ are convex, then Problem $\mathrm{LP}_{\bar{f},\bar{g}}$ defining our baselines can be equivalently re-written by using strategies $\boldsymbol{x} \in \mathcal{X}$ rather than strategy mixtures $\boldsymbol{\xi} \in \Xi$, since there always exists an optimal solution to Problem $\mathrm{LP}_{\bar{f},\bar{g}}$ that places all the probability mass on a single strategy.

Our goal is to design online algorithms for the decision maker that output a sequence of decisions $\boldsymbol{x}_1, \ldots, \boldsymbol{x}_T$ such that both the *cumulative regret* with respect to the performance of the baseline, defined as $R^T \coloneqq T\,\mathrm{OPT}_{\bar{f},\bar{g}} - \sum_{t=1}^{T} f_t(\boldsymbol{x}_t)$, and the *cumulative constraints violation*, defined as $V^T \coloneqq \max_{i \in [m]} \sum_{t=1}^{T} g_{t,i}(\boldsymbol{x}_t)$, grow sublinearly in the number of rounds $T$.

In conclusion, we introduce a problem-specific parameter that is strictly related to the feasibility of Problem $\mathrm{LP}_{\bar{f},\bar{g}}$. We call it the *feasibility parameter* $\rho \in \mathbb{R}$, which is formally defined as follows:

- When functions $g_t$ are selected stochastically, $\rho \coloneqq \sup_{\boldsymbol{\xi} \in \Xi} \min_{i \in [m]} -\bar{g}_i(\boldsymbol{\xi})$.

- When functions $g_t$ are selected adversarially, $\rho \coloneqq \sup_{\boldsymbol{\xi} \in \Xi} \min_{t \in [T]} \min_{i \in [m]} -g_{t,i}(\boldsymbol{\xi})$.

Intuitively, in the stochastic case, $\rho$ is equal to $d_{\bar{g}}$, while in the adversarial case it is computed similarly, but considering the worst case with respect to the functions $g_t$ observed at each round $t$. Notice that, when $\rho > 0$, Slater's condition is satisfied for Problem $\mathrm{LP}_{\bar{f},\bar{g}}$.

In the following, we denote by $\boldsymbol{\xi}^* \in \Xi$ a strategy mixture that is optimal for Problem $\mathrm{LP}_{\bar{f},\bar{g}}$. Moreover, we always assume that functions $f_t$ and $g_t$ are such that Problem $\mathrm{LP}_{\bar{f},\bar{g}}$ is feasible, and we let $\boldsymbol{\xi}^\circ \in \Xi$ be the *feasible* strategy mixture that is optimal for the problem defining $\rho$.[5]

## 2.3 Regret minimizers

A *regret minimizer* (RM) for a set $\mathcal{W}$ is an abstract model for a decision maker repeatedly interacting with an environment. At each $t$, a RM performs two operations: (i) NEXTELEMENT(), which outputs an element $\boldsymbol{w}_t \in \mathcal{W}$; and (ii) OBSERVEUTILITY($\cdot$), which updates the internal state of the RM using the feedback received from the environment. This is either a utility function $u_t : \mathcal{W} \to [a, b]$ having range $[a, b] \subseteq \mathbb{R}$ (*full feedback*) or only the value $u_t(\boldsymbol{w}_t)$ (*bandit feedback*), with $u_t$ possibly depending adversarially on $\boldsymbol{w}_1, \ldots, \boldsymbol{w}_{t-1}$. The objective of the RM is to output a sequence $\boldsymbol{w}_1, \ldots, \boldsymbol{w}_T$ of points in $\mathcal{W}$ so that its *cumulative regret*, defined as $\sup_{\boldsymbol{w} \in \mathcal{W}} \sum_{t=1}^{T} (u_t(\boldsymbol{w}) - u_t(\boldsymbol{w}_t))$, grows asymptotically sublinearly in $T$. See [19] for a review of the various RMs available in the literature.

For the ease of presentation, we introduce the concept of *regret minimizer constructor*, which is a procedure, say INIT($\mathcal{W}, [a, b], \eta$), that builds a RM on the basis of the three parameters given as input. In particular, the procedure returns a RM instantiated for the set $\mathcal{W}$, working with utility functions having range $[a, b]$, and such that its cumulative regret is guaranteed to grow sublinearly in the time horizon $T$ with probability at least $1 - \eta$.

## 3 A unifying meta-algorithm

In this section, we present our meta-algorithm. Its core idea is to instantiate suitable pairs of RMs, where one is working in the domain $\mathcal{X}$ of primal variables and the other in a suitable subset of the domain $\mathbb{R}^m_+$ of dual variables. At each round $t \in [T]$, the algorithm makes the RMs "play" against each other in a *Lagrangian game*, where the utility functions observed by them are related to the Lagrangian function $\mathcal{L}_{f_t,g_t}(\boldsymbol{x}, \boldsymbol{\lambda})$ of Problem $\mathrm{LP}_{f_t,g_t}$.[6]

Algorithm 1 provides the pseudo-code of the meta-algorithm, which takes as input: the total number of rounds $T$, a failure probability $\delta \in (0, 1)$ such that the guarantees provided by the algorithm hold with probability at least $1 - \delta$, and a lower bound $\hat{\rho} \geq 0$ on the value of the feasibility parameter $\rho$.

**Algorithm description.** The algorithm works in two phases. In the first one, called *play phase*, the algorithm builds a primal RM, called $\mathcal{R}^{\mathrm{P}}_{\mathrm{I}}$, working in the primal domain $\mathcal{X}$ and a dual RM, called $\mathcal{R}^{\mathrm{D}}_{\mathrm{I}}$, operating on the subset $\mathcal{D}_{\tilde{\rho}}$ of the dual domain $\mathbb{R}^m_+$, where $\tilde{\rho}$ is set in Line 1. The algorithm makes the two RMs playing against each other (see the call LAGRANGIANGAME($\mathcal{R}^{\mathrm{P}}_{\mathrm{I}}, \mathcal{R}^{\mathrm{D}}_{\mathrm{I}}, 1$)) until either the cumulative violation $V^t$ incurred by the algorithm exceeds a given threshold (see Line 4, where $M_{\tilde{\rho}}$ is defined in Equation (2)) or round $T$ is reached. Then, in the second phase, called *recovery phase*, the algorithm constructs a new pair of primal, dual RMs, with the latter working on the $(m-1)$-dimensional simplex $\Delta_m$. The recovery phase uses the remaining rounds to make these new RMs play against each other, with the primal RM observing modified utility functions that do *not* account for functions $f_t$ (see the call LAGRANGIANGAME($\mathcal{R}^{\mathrm{P}}_{\mathrm{II}}, \mathcal{R}^{\mathrm{D}}_{\mathrm{II}}, 0$)). Intuitively, this is needed in order to ensure that the algorithm plays strategies $\boldsymbol{x}_t$ that satisfy the constraints, thus balancing out the cumulative constraint violation accumulated in the first phase. The pseudo-code describing one "play" between two RMs, called $\mathcal{R}^{\mathrm{P}}$ and $\mathcal{R}^{\mathrm{D}}$, is defined by the sub-procedure LAGRANGIANGAME($\mathcal{R}^{\mathrm{P}}, \mathcal{R}^{\mathrm{D}}, v$) in Algorithm 2. The additional parameter $v \in \{0, 1\}$ is used to control the feedback fed into the primal RM $\mathcal{R}^{\mathrm{P}}$; specifically, if $v = 1$, then $\mathcal{R}^{\mathrm{P}}$ observes a utility function that also accounts for $f_t$ (play phase), otherwise, if $v = 0$, the observed utility function only accounts for the term depending on $g_t$ (recovery phase).

**Regret minimizer constructors.** Algorithm 1 also needs access to two suitably-defined regret minimizer constructors, namely INIT$^{\mathrm{P}}(\mathcal{W}, [a, b], \eta)$ and INIT$^{\mathrm{D}}(\mathcal{W}, [a, b], \eta)$, where the former is used

---

[5]Notice that $\boldsymbol{\xi}^*$ and $\boldsymbol{\xi}^\circ$ may *not* be well defined in all the cases in which the problem that defines them does *not* admit a maximum. Nevertheless, in such cases, we assume that $\boldsymbol{\xi}^*$ (or $\boldsymbol{\xi}^\circ$) is a strategy mixture arbitrarily "close" to the supremum, so that all of our results continue to hold up to negligible additive approximations that are dominated by other approximation factors, and we can safely ignore them for ease of exposition.

[6]The idea of having pairs of primal, dual RMs playing a Lagrangian game was originally introduced by Immorlica et al. [30], restricted to the case of knapsack constraints.

| **Algorithm 1** META-ALGORITHM$(T, \delta, \hat{\rho})$ | **Algorithm 2** LAGRANGIANGAME$(\mathcal{R}^{\mathrm{P}}, \mathcal{R}^{\mathrm{D}}, v)$ |
|---|---|
| 1: $\tilde{\rho} \leftarrow \max\left\{\hat{\rho}/2, T^{-1/4}\right\}, \eta \leftarrow \delta/3, t \leftarrow 1$ | 1: $\boldsymbol{x}_t \leftarrow \mathcal{R}^{\mathrm{P}}.\text{NEXTELEMENT}()$ |
| ▷ Phase I: Play | 2: $\boldsymbol{\lambda}_t \leftarrow \mathcal{R}^{\mathrm{D}}.\text{NEXTELEMENT}()$ |
| 2: $\mathcal{R}_{\mathrm{I}}^{\mathrm{P}} \leftarrow \text{INIT}^{\mathrm{P}}\left(\mathcal{X}, \left[-1/\tilde{\rho}, 1+1/\tilde{\rho}\right], \eta\right)$ | 3: $\quad$ Play $\boldsymbol{x}_t$ and get $f_t$ and $g_t$ $\qquad$ ▷ Full f. |
| 3: $\mathcal{R}_{\mathrm{I}}^{\mathrm{D}} \leftarrow \text{INIT}^{\mathrm{D}}\left(\mathcal{D}_{\tilde{\rho}}, \left[-1/\tilde{\rho}, 1/\tilde{\rho}\right], 0\right)$ | $\quad$ Play $\boldsymbol{x}_t$ and get $f_t(\boldsymbol{x}_t)$ and $g_t(\boldsymbol{x}_t)$ ▷ Bandit f. |
| 4: **while** $V^t \leq (T-t)\tilde{\rho} + M_{\tilde{\rho}} - 1 \wedge t \leq T$ **do** | ▷ Primal RM update |
| 5: $\quad \boldsymbol{x}_t \leftarrow \text{LAGRANGIANGAME}(\mathcal{R}_{\mathrm{I}}^{\mathrm{P}}, \mathcal{R}_{\mathrm{I}}^{\mathrm{D}}, 1)$ | 4: $\quad$ Let $u_t^{\mathrm{P}} : \boldsymbol{x} \mapsto v f_t(\boldsymbol{x}) - \langle \boldsymbol{\lambda}_t, g_t(\boldsymbol{x}) \rangle$ $\quad$ ▷ Full f. |
| 6: $\quad t \leftarrow t + 1$ | $\quad u_t^{\mathrm{P}}(\boldsymbol{x}_t) \leftarrow v f_t(\boldsymbol{x}_t) - \langle \boldsymbol{\lambda}_t, g_t(\boldsymbol{x}_t) \rangle$ ▷ Bandit f. |
| 7: $T_1 \leftarrow t - 1$ | 5: $\quad \mathcal{R}^{\mathrm{P}}.\text{OBSERVEUTILITY}(u_t^{\mathrm{P}})$ $\qquad$ ▷ Full f. |
| ▷ Phase II: Recovery | $\quad \mathcal{R}^{\mathrm{P}}.\text{OBSERVEUTILITY}(u_t^{\mathrm{P}}(\boldsymbol{x}_t))$ $\quad$ ▷ Bandit f. |
| 8: $\mathcal{R}_{\mathrm{II}}^{\mathrm{P}} \leftarrow \text{INIT}^{\mathrm{P}}\left(\mathcal{X}, [-1, 1], \eta\right)$ | ▷ Dual RM update |
| 9: $\mathcal{R}_{\mathrm{II}}^{\mathrm{D}} \leftarrow \text{INIT}^{\mathrm{D}}\left(\Delta_m, [-1, 1], 0\right)$ | 6: Let $u_t^{\mathrm{D}} : \boldsymbol{\lambda} \mapsto \langle \boldsymbol{\lambda}, g_t(\boldsymbol{x}) \rangle$ |
| 10: **while** $t \leq T$ **do** | 7: $\mathcal{R}^{\mathrm{D}}.\text{OBSERVEUTILITY}(u_t^{\mathrm{D}})$ |
| 11: $\quad \boldsymbol{x}_t \leftarrow \text{LAGRANGIANGAME}(\mathcal{R}_{\mathrm{II}}^{\mathrm{P}}, \mathcal{R}_{\mathrm{II}}^{\mathrm{D}}, 0)$ | |
| 12: $\quad t \leftarrow t + 1$ | |

to build RMs working in the primal domain and the latter for those operating on the dual one. Their actual implementation depends on the specific problem at hand. In the following, we let $\mathcal{E}_{t,\eta}^{\mathrm{P}}$ be the regret upper bound (on $t \in [T]$ rounds) for primal RMs $\mathcal{R}^{\mathrm{P}}$ dealing with utility functions having range $[0, 1]$, as returned by the call $\text{INIT}^{\mathrm{P}}(\mathcal{X}, [0, 1], \eta)$. Notice that, when the range is $[a, b]$, the same RM can be adopted by first normalizing utility values, so that the resulting regret upper bound is $(b - a)\mathcal{E}_{t,\eta}^{\mathrm{P}}$. As for dual RMs $\mathcal{R}^{\mathrm{D}}$, we let $\mathcal{E}_t^{\mathrm{D}}$ be the regret upper bound (on $t \in [T]$ rounds) provided by the RM defined for the set $\Delta_m$, while $\mathcal{E}_t^{\mathrm{D}}/\tilde{\rho}$ is the upper bound for the dual RM instantiated on the set $\mathcal{D}_{\tilde{\rho}}$. Notice that, since dual RMs always have full feedback, we can safely assume that the regret bounds $\mathcal{E}_t^{\mathrm{D}}$ hold deterministically. We also assume that RMs provide bounds that increase with the number of rounds, *i.e.*, such that $\mathcal{E}_{t,\eta}^{\mathrm{P}} \leq \mathcal{E}_{t',\eta}^{\mathrm{P}}$ and $\mathcal{E}_t^{\mathrm{D}} \leq \mathcal{E}_{t'}^{\mathrm{D}}$ for all $t \leq t'$.

**How to construct RMs.** $\text{INIT}^{\mathrm{D}}$ can be implemented by using *online mirror descent* (OMD) with domain $\Delta_m$ (or $\mathcal{D}_1$) and a negative entropy regularizer. Since the utility function $u_t^{\mathrm{D}}$ is linear in $\boldsymbol{\lambda}$, we get a regret bound for the primal RM of $\mathcal{E}_T^{\mathrm{D}} = O(\sqrt{T \log(m)})$ (see, *e.g.*, [12, 37]). The design of $\text{INIT}^{\mathrm{P}}$ depends on the structure of $\mathcal{X}$ and functions $f_t$ and $g_t$. For instance, in convex settings with full feedback we can employ OMD [29], while with bandit feedback we can use [15]. Finally, for non-convex functions we can employ, *e.g.*, the RMs in [39]. All these RMs guarantee $\tilde{O}(\sqrt{T})$ regret.

**How to get away with no knowledge of $\rho$.** In Section 7, we show that a lower bound $\hat{\rho}$ is *not* necessary when functions $g_t$ are selected stochastically. Indeed, it is sufficient to add a preliminary phase to Algorithm 1, which is used to infer a suitable lower bound on $\rho$ from experience. In order to do this, only $\sqrt{T}$ rounds are needed, so that the bounds on the cumulative regret and the cumulative constraint violation achieved by the algorithm are *not* compromised. When functions $g_t$ are chosen adversarially, as we show in Section 7, it is impossible to compute a lower bound on the feasibility parameter $\rho$ by only adding a preliminary phase to Algorithm 1 which uses $\sqrt{T}$ rounds.

**Remark 3.1** (Dependence on the lower bound $\hat{\rho}$). *Algorithm 1 can take as input any $\hat{\rho} \geq 0$. However, since our regret bounds include a factor $1/\tilde{\rho}$, by choosing the trivial lower bound $\hat{\rho} = 0$ we incur in a regret of $\tilde{O}(\sqrt{T}/\tilde{\rho}) = \tilde{O}(T^{3/4})$. In order to obtain optimal bounds, we would like to have $\tilde{\rho} = \Omega(\rho)$.*

**Remark 3.2** (Dependence on the feasibility parameter $\rho$). *In our analysis, we include the dependence on the feasibility parameter $\rho$ in the regret and constraint violation bounds achieved by the algorithm. As customary, the goal is devising bounds of the form* poly(instance) $\cdot h(T)$, *where the first term is a polynomial function of the parameters defining the problem instance, and $h(T) = o(T)$. Therefore, we cannot include a factor $1/\rho$ in the regret bounds if $\rho$ can be arbitrarily small. Even from a practical standpoint, when $\rho$ is too small, a $1/\rho$ regret bound is too large to be significant. For those reasons, we set $\tilde{\rho}$ in Algorithm 1 to be the maximum between the feasibility parameter lower bound $\hat{\rho}$ and $T^{-1/4}$. Notice that the value $T^{-1/4}$ has been carefully chosen so as to minimize the maximum between the cumulative regret and the cumulative constraint violation when the lower bound on the feasibility parameter $\hat{\rho}$ is too small.*

# 4 Analysis with stochastic constraints and adversarial rewards

We start by analyzing the performance of our meta-algorithm (Algorithm 1) when the reward and constraint functions are selected stochastically and adversarially, respectively.

Given $t \in [T]$ and $\eta \in (0, 1)$, we let $\mathcal{E}_{t,\eta} \coloneqq \sqrt{8t \log(18mt^2/\eta)}$ be the value bounding differences between expectations and empirical means of constraint functions, obtained by applying the Azuma-Hoeffding inequality, and holding with probability at least $1 - \eta$. Given $\gamma \in (0, 1)$, we also let

$$M_\gamma \coloneqq \frac{2}{\gamma}\sqrt{T} + \left(2 + \frac{3}{\gamma}\right)\mathcal{E}_{t,\eta} + \left(1 + \frac{2}{\gamma}\right)\mathcal{E}_{t,\eta}^{\mathrm{P}} + \frac{1}{\gamma}\mathcal{E}_t^{\mathrm{D}}, \tag{2}$$

which is a recurring term related to the maximum violation that Algorithm 1 accepts in play phase.

First, we introduce a useful event $\mathbf{E}$ that encompasses all the cases in which Algorithm 1 successfully terminates. Then, Lemma 4.2 shows that such an event holds with probability at least $1 - \delta$. In particular, $\mathbf{E}$ holds when the regret bounds of $\mathcal{R}_{\mathrm{I}}^{\mathrm{P}}$ and $\mathcal{R}_{\mathrm{II}}^{\mathrm{P}}$ hold, and, additionally, the differences between expectations and empirical means of constraint functions are bounded as desired.

**Definition 4.1.** We denote with $\mathbf{E}$ the event in which Algorithm 1 satisfies the following conditions (recall that $\eta = \delta/3$): (i) the regret incurred by $\mathcal{R}_{\mathrm{I}}^{\mathrm{P}}$ after $T_1$ rounds is upper bounded by $\mathcal{E}_{T_1,\eta}^{\mathrm{P}}$; (ii) the regret cumulated by $\mathcal{R}_{\mathrm{II}}^{\mathrm{P}}$ after the remaining $T - T_1$ rounds is upper bounded by $\mathcal{E}_{T-T_1,\eta}^{\mathrm{P}}$; and (iii) for every pair of rounds $t, t' \in [T] : t \leq t'$ and resource $i \in [m]$ it holds:

- $\left|\sum_{\tau=t}^{t'} g_{\tau,i}(\boldsymbol{x}_\tau) - \sum_{\tau=t}^{t'} \bar{g}_i(\boldsymbol{x}_\tau)\right| \leq \mathcal{E}_{t'-t,\eta}$,
- $\left|\sum_{\tau=t}^{t'} \boldsymbol{\lambda}_\tau g_{\tau,i}(\boldsymbol{x}_\tau) - \sum_{\tau=t}^{t'} \boldsymbol{\lambda}_\tau \bar{g}_i(\boldsymbol{x}_\tau)\right| \leq \mathcal{E}_{t'-t,\eta} \max_{\tau \in [T]:t \leq \tau \leq t'} \|\boldsymbol{\lambda}_\tau\|_1$,
- $\left|\sum_{\tau=t}^{t'} g_{\tau,i}(\boldsymbol{\xi}) - \sum_{\tau=t}^{t'} \bar{g}_i(\boldsymbol{\xi})\right| \leq \mathcal{E}_{t'-t,\eta}$ for $\boldsymbol{\xi} \in \{\boldsymbol{\xi}^*, \boldsymbol{\xi}^\circ\}$,
- $\left|\sum_{\tau=t}^{t'} \boldsymbol{\lambda}_\tau g_{\tau,i}(\boldsymbol{\xi}) - \sum_{\tau=t}^{t'} \boldsymbol{\lambda}_\tau \bar{g}_i(\boldsymbol{\xi})\right| \leq \mathcal{E}_{t'-t,\eta} \max_{\tau \in [T]:t \leq \tau \leq t'} \|\boldsymbol{\lambda}_\tau\|_1$ for $\boldsymbol{\xi} \in \{\boldsymbol{\xi}^*, \boldsymbol{\xi}^\circ\}$.

**Lemma 4.2.** *After running Algorithm 1, the event $\mathbf{E}$ holds with probability at least $1 - \delta$.*

Next, we lower bound the cumulative reward obtained by Algorithm 1 during the play phase. Intuitively, we show that, if the cumulative constraints violation is large, then the decisions $\boldsymbol{x}_t$ in the first $T_1$ rounds provide a per-round reward much higher than that achievable by $\boldsymbol{\xi}^*$. This allows us to employ the following recovery phase to decrease constraints violation cumulated in the play phase, while also ensuring that the cumulative regret stays low at the end of the algorithm. Formally:

**Lemma 4.3.** *If event $\mathbf{E}$ holds, then after round $T_1$ of Algorithm 1 the following inequality holds:* $\sum_{t=1}^{T_1} f_t(\boldsymbol{x}_t) \geq \sum_{t=1}^{T_1} f_t(\boldsymbol{\xi}^*) + (T - T_1) - \frac{1}{\rho}\mathcal{E}_{T_1,\eta} - \left(1 + \frac{2}{\rho}\right)\mathcal{E}_{T_1,\eta}^{\mathrm{P}} - \frac{1}{\rho}\mathcal{E}_{T_1}^{\mathrm{D}}.$

In the recovery phase, the only goal of Algorithm 1 is to decrease constraints violation. In the following Lemma 4.4, we show that, at each round of the recovery phase, the algorithm is "close" to satisfying (in expectation) all the constraints by at least $\rho$. Formally:

**Lemma 4.4.** *If event $\mathbf{E}$ holds, then after Algorithm 1 halts, the following holds for every $i \in [m]$:* $\sum_{t=T_1+1}^{T} g_{t,i}(\boldsymbol{x}_t) \leq -(T - T_1)\rho + 2\mathcal{E}_{T-T_1,\eta}^{\mathrm{P}} + \mathcal{E}_{T-T_1}^{\mathrm{D}} + \mathcal{E}_{T-T_1,\eta}.$

Now, we are ready to present the two main results of this section. First, we provide a bound on the cumulative regret and constraints violation when the lower bound $\hat{\rho}$ is sufficiently large.

**Condition 4.5.** *It holds that $\hat{\rho} \geq 2T^{-1/4}$.*

Notice that, under Condition 4.5, $\tilde{\rho} = \hat{\rho}/2$. This gives us the following result:

**Theorem 4.6.** *Suppose that functions $f_t$ and $g_t$ are selected adversarially and stochastically, respectively. If Condition 4.5 is satisfied, then, with probability at least $1 - \delta$, Algorithm 1 provides* $R^T \leq \frac{1}{\rho}\mathcal{E}_{T,\eta} + \left(1 + \frac{2}{\rho}\right)\mathcal{E}_{T,\eta}^{\mathrm{P}} + \frac{1}{\rho}\mathcal{E}_T^{\mathrm{D}}$ *and* $V^T \leq M_{\tilde{\rho}} + 2\mathcal{E}_{T,\eta}^{\mathrm{P}} + \mathcal{E}_T^{\mathrm{D}} + \mathcal{E}_{T,\eta}.$

Finally, we also prove that even if Condition 4.5 is *not* satisfied, *i.e.*, the lower bound $\hat{\rho}$ is *not* sufficiently large, the following holds:

**Theorem 4.7.** *Suppose that functions $f_t$ and $g_t$ are selected adversarially and stochastically, respectively. Algorithm 1 guarantees that the following bounds hold with probability at least $1 - \delta$:*
$R_T \leq T^{1/4}\mathcal{E}_{T,\eta} + \left(1 + 2T^{1/4}\right)\mathcal{E}_{T,\eta}^{\mathbb{P}} + T^{1/4}\mathcal{E}_T^{\mathbb{D}}$ *and* $V_T \leq T^{3/4} + M_{T^{-1/4}} + 2\mathcal{E}_{T,\eta}^{\mathbb{P}} + \mathcal{E}_T^{\mathbb{D}} + \mathcal{E}_{T,\eta}$.

**Remark 4.8.** *Notice that, by using primal and dual RMs whose regret bounds are of the order of $\tilde{O}(\sqrt{T})$, Theorem 4.6 allows us to recover $\tilde{O}(\sqrt{T}/\hat{\rho})$ regret and $\tilde{O}(\sqrt{T}/\hat{\rho})$ constraints violation for the case in which Condition 4.5 holds. Theorem 4.7 still provides $\tilde{O}(T^{3/4})$ regret and constraints violation when the condition is not met, which is necessary the case when $\rho = 0$.*

# 5 Analysis with stochastic constraints and stochastic rewards

In this section, we focus on the case in which both reward and constraint functions are selected stochastically. In this setting, we are able to show that Algorithm 1 never enters the recovery phase. As we argue in Section 8, this is an important property for budget-management applications, since it is related to the round in which the budget is fully depleted.

In order to prove our result, we extend the event **E** to capture also the Azuma-Hoeffding bounds for the reward functions, which are stochastic in this setting.[7] The core idea that we exploit to prove our result is that we can think of the two RMs as if they are playing a stochastic repeated zero-sum game, which is the repeated Lagrangian game whose functions are sampled according to the probability measures $\mu_{\mathcal{F}}$ and $\mu_{\mathcal{G}}$. By Theorem 2.3, strong duality holds, and the game has an equilibrium. Hence, it is possible to show that the per-round utility of the primal RM is close to the value of the game, which is $\mathrm{OPT}_{\bar{f},\bar{g}}$. At the same time, it is possible to show that, if the cumulative constraints violation becomes large during the play phase (and, thus, $T_1 < T$), then the per-round utility of the primal RM is below $\mathrm{OPT}_{\bar{f},\bar{g}}$, reaching a contradiction that proves the following theorem.

**Theorem 5.1.** *Suppose that functions $f_t$ and $g_t$ are selected stochastically. With probability at least $1 - \delta$, Algorithm 1 never enters the recovery phase, namely $T_1 = T$.*

Notice that regret bounds analogous to the ones in Theorems 4.6 and 4.7 also hold in the case in which both reward and constraint functions are selected stochastically.

# 6 Analysis with adversarial constraints

In this section, we study settings in which the constraint functions $g_t$ are selected adversarially. As shown by Mannor et al. [36], it is impossible to obtain sublinear cumulative regret and constraints violation when using our baseline, *i.e.*, the best fixed strategy mixture $\boldsymbol{\xi}^*$ satisfying (in expectation) the long-tern constraints. However, we show that it is possible to achieve a $\rho/(1 + \rho)$ fraction of the cumulative reward obtained by always playing $\boldsymbol{\xi}^*$, while guaranteeing sublinear constraints violation. The dependence of the approximation factor on the feasibility parameter $\rho$ is similar to the dependence on the per-round budget in problems with budget constraints (see the related works in Appendix A for more details). Moreover, as we discuss later in Section 8, when restricted to the case of budget constraints and adversarial reward/cost functions, our approximation factor matches the state-of-the-art bounds provided by Castiglioni et al. [17].

As a first step to prove our result, we provide a lower bound on the cumulative reward of the primal RM during the play phase. We show that it achieves at least a $\rho/(1 + \rho)$ fraction of the value obtained by the optimal solution in the first $T_1$ rounds. Moreover, the algorithm provides an additional utility compensating for the last rounds in which the algorithm only focuses in satisfying the constraints. Finally, we show that, in the recovery phase, the constraints are satisfied by at least $\rho$ at each round, up to a term related to the regret of $\mathcal{R}_{\mathrm{II}}^{\mathbb{P}}$ and $\mathcal{R}_{\mathrm{II}}^{\mathbb{D}}$, proving the following theorem.

**Theorem 6.1.** *Suppose that functions $f_t$ and $g_t$ are selected adversarially. If Condition 4.5 is satisfied, then, with probability at least $1 - \frac{2}{3}\delta$, Algorithm 1 guarantees that the following holds:*
$\sum_{t=1}^{T} f_t(\boldsymbol{x}_t) \geq \frac{\rho}{1+\rho}\sum_{t=1}^{T}\mathrm{OPT}_{\bar{f},\bar{g}} - \left(1 + \frac{2}{\rho}\right)\mathcal{E}_{T,\eta}^{\mathbb{P}} - \frac{1}{\rho}\mathcal{E}_T^{\mathbb{D}}$ *and* $V^T \leq M_{\tilde{\rho}} + 2\mathcal{E}_{T,\eta}^{\mathbb{P}} + \mathcal{E}_T^{\mathbb{D}}$.

A similar result can be also derived for the case of stochastic rewards and adversarial constraints.

---

[7]Accounting for the martingale difference sequences $f_t(\boldsymbol{x}_t) - \bar{f}(\boldsymbol{x}_t)$ and $f_t(\boldsymbol{\xi}^*) - \bar{f}(\boldsymbol{\xi}^*)$.

**Corollary 6.2.** *Suppose functions $f_t$ and $g_t$ are selected stochastically and adversarially, respectively. If Condition 4.5 is satisfied, then, with probability at least $1 - \delta$, Algorithm 1 provides $\sum_{t=1}^{T} f_t(\boldsymbol{x}_t) \geq \frac{\rho}{1+\rho} \sum_{t=1}^{T} \text{OPT}_{\bar{f}, \bar{g}} - \left(1 + \frac{2}{\bar{\rho}}\right) \mathcal{E}_{T,\eta}^{\text{P}} - \frac{1}{\bar{\rho}} \mathcal{E}_{T}^{\text{D}} - 2\mathcal{E}_{T,\eta}$ and $V^T \leq M_{\tilde{\rho}} + 2\mathcal{E}_{T,\eta}^{\text{P}} + \mathcal{E}_{T}^{\text{D}} + \mathcal{E}_{T,\eta}$.*

**Remark 6.3.** *By using primal and dual RMs whose regret bounds are of the order of $\tilde{O}(\sqrt{T})$, Theorem 6.1 and Corollary 6.2 allows us to recover $\sum_{t=1}^{T} f_t(\boldsymbol{x}_t) \geq \frac{\rho}{1+\rho} \sum_{t=1}^{T} \text{OPT}_{\bar{f}, \bar{g}} - \tilde{O}(\sqrt{T}/\hat{\rho})$, and $\tilde{O}(\sqrt{T}/\hat{\rho})$ constraints violation for the case in which Condition 4.5 holds.*

## 7 How to get away with no knowledge about the feasibility parameter

We show how to extend Algorithm 1 in order to deal with settings in which a lower bound on the feasibility parameter $\rho$ is *not* known, when functions $g_t$ are selected stochastically. We propose Algorithm 3, which runs Algorithm 1 by first devoting a given number $T_0 < T$ of rounds to inferring a suitable lower bound $\hat{\rho}$ on the feasibility parameter $\rho$. Ideally, we would like to have $\hat{\rho} = \Omega(\rho)$, so as to recover bounds of the order $\tilde{O}(\sqrt{T}/\rho)$. In particular, we show that we can run Algorithm 3 with $T_0 = T^{1/2}$ to obtain an approximation of $\rho$ that has an additive approximation error of the order $T^{1/4}$. This is sufficient to get $\hat{\rho} = \Omega(\rho)$, since a good approximation of $\rho$ is only needed when $\rho \geq T^{1/4}$.

Let us remark that our approach only works when constraints functions $g_t$ are selected stochastically. When these are chosen adversarially, it is easy to see that it is impossible to compute a lower bound on the feasibility parameter $\rho$ by only using the first rounds. For instance, think of a setting in which $\rho$ is very large in the first rounds, while it becomes small during the last ones.

To exploit the guarantees of Algorithm 1 presented in the previous sections, it is enough to show that, after the first $T_0$ rounds of Algorithm 3, $\hat{\rho} \leq \rho$ holds with high probability.

---

**Algorithm 3** META-ALGORITHM$(T, T_0, \delta)$

1:  $\mathcal{R}^{\text{P}} \leftarrow \text{INIT}^{\text{P}}\left(\mathcal{X}, \left[-1, 1\right], \delta\right)$
2:  $\mathcal{R}^{\text{D}} \leftarrow \text{INIT}^{\text{D}}\left(\Delta_m, [-1, 1], 0\right)$
3:  $t \leftarrow 1$
4:  **while** $t \leq T_0$: **do**
5:  $\quad \boldsymbol{x}_t \leftarrow \text{LAGRANGIANGAME}(\mathcal{R}^{\text{P}}, \mathcal{R}^{\text{D}}, 0)$
6:  $\quad t \leftarrow t + 1$
7:  $\hat{\rho} \leftarrow -\frac{1}{T_0}\left(\max_{i \in [m]} \sum_{t=1}^{T_0} g_{t,i}(\boldsymbol{x}_t) + \mathcal{E}_{T_0, \delta}\right)$
8:  Run Algorithm 1 with $T - T_0$, $\delta$, and $\hat{\rho}$ as inputs

---

**Lemma 7.1.** *If $T_0 = \sqrt{T}$, then Algorithm 3 guarantees that $\hat{\rho} \leq \rho$ with probability at least $1 - \delta$.*

In order to recover a good estimate of $\rho$, we need the value of $\rho$ to be sufficiently large.

**Condition 7.2.** *It holds that $\rho \geq \frac{2}{T_0}(2\mathcal{E}_{T_0, \delta} + 2\mathcal{E}_{T_0, \delta}^{\text{P}} + \mathcal{E}_{T_0}^{\text{D}})$.*

**Remark 7.3.** *Notice that, by using primal and dual RMs whose regret bounds are of the order $\tilde{O}(\sqrt{T})$ and by setting $T_0 = \sqrt{T}$, Condition 7.2 is satisfied when $\rho = \omega(T^{-1/4})$.*

Next, we show that $\hat{\rho} = \Omega(\rho)$, which allows us to exploit the guarantees proved for Algorithm 1 in order to provide analogous ones for Algorithm 3. Formally:

**Lemma 7.4.** *By setting $T_0 = \sqrt{T}$, and assuming that Condition 7.2 is satisfied, after $T_0$ rounds of Algorithm 3 we have that $\hat{\rho} \geq \rho/2$ with probability at least $1 - 2\delta$.*

By applying the results of the previous sections on the guarantees of Algorithm 1, and by using primal and dual RMs whose regret bounds are of the order $\tilde{O}(\sqrt{T})$, we get $\tilde{O}(\sqrt{T}/\rho)$ and $\tilde{O}(\sqrt{T}/\rho)$ regret and violation bounds, respectively, when the functions $g_t$ are selected stochastically.

## 8 Applications to repeated auctions settings

Internet advertising platforms usually operationalize large auction markets by using *proxy bidders* that place bids in repeated auctions on the advertisers' behalf. A proxy-bidder selects bids according to a *budget-pacing mechanism*, which manages the usage of the advertisers' budget over time [1, 22, 9]. In this section, we discuss the application of our framework to budget-management in auctions, arguing that it can deal with more general constraints on ad slots allocation with respect to what is currently achievable with multiplicative pacing algorithms, which manage only *knapsack constraints*.

We consider the problem faced by a bidder who takes part in a sequence of repeated auctions. We focus on the case of *second-price* and *first-price* auctions, since they are the *de facto* standard in large Internet advertising platforms. At each round $t \in [T]$, the bidder observes their valuation $v_t$ from a finite set of $n_v$ possible valuations $\mathcal{V} \subset [0, 1]$. Such valuation models targeting preferences of the advertiser. Then, the bidder chooses a bid $b_t \in \mathcal{B}$, where $\mathcal{B} \subset [0, 1]$ is a finite set of $n_b$ possible bids such that $0 \in \mathcal{B}$ (*i.e.*, the bidder is allowed to skip items without incurring in any cost). The utility of the bidder depends on the largest among competing bids, denoted by $\beta_t$. In particular, the utility is computed as $f_t(b_t) = (v_t - c_t(b_t))\mathbb{1}\{b_t \geq \beta_t\}$, where the cost $c_t$ is such that $c_t(b_t) = \beta\mathbb{1}\{b_t \geq \beta_t\}$ in second-price auctions, and $c_t(b_t) = b_t\mathbb{1}\{b_t \geq \beta_t\}$ for first-price ones. Finally, the bidder has a target *per-round* budget of $\rho > 0$, which yields an overall budget $B := \rho T$ that limits the total spending over the $T$ rounds. In the case of budget-constrained bidding, a strictly feasible solution can be easily achieved by always bidding 0. Using the target per-round budget $\rho = B/T$ we can write the budget constraint as $\sum_{t \in [T]} g_t(b_t) \leq 0$, with $g_t(b) = c_t(b) - \rho$ for any $b \in \mathcal{B}$. Notice that, in this setting, we have the same feasibility parameter $\rho$ for both the stochastic and the adversarial case.

As a benchmark to evaluate the algorithm, we consider the best feasible static policy $\pi : \mathcal{V} \to \mathcal{B}$. The set of static policies can be represented by $\mathcal{X} := \mathcal{B}^{n_v}$, where a vector $\boldsymbol{b} \in \mathcal{B}^{n_v}$ encodes the policy's bids for each possible valuation. To apply our framework to this problem, it is sufficient to design a primal regret minimizer constructor (recall that, in order to design dual RMs, we can employ OMD). This can be implemented by instantiating a regret minimizer EXP3.P [5] for each possible valuation in $\mathcal{V}$. Given a failure probability $\nu \in (0, 1)$, each RM guarantees a regret bound $O(\sqrt{Tn_b \log(n_b/\nu)})$ with probability at least $1 - \nu$. Thus, given a desired failure probability $\eta \in (0, 1)$, by setting $\nu = \eta/n_v$ we get that, with probability at least $1 - \eta$, the bounds of all the RMs hold. Hence, by a union bound, we get that the regret of a primal RM is $\mathcal{E}_{T,\eta}^{\mathbb{p}} = O(n_v \sqrt{Tn_b \log(n_b n_v/\eta)})$.

**Guaranteed budget completion in the stochastic case.**   The crux of budget-pacing mechanisms is ensuring that the advertisers' budget is not depleted too early (thereby missing potentially valuable future advertising opportunities), while being fully depleted within the planned duration of the campaign. Theorem 5.1 shows that, when inputs are generated according to some stochastic model, Algorithm 1 never enters the recovery phase. This is crucial in the context of budget-pacing mechanisms, because whenever the algorithm enters the recovery phase it will converge to always bid 0 in order to mitigate constraints violation. Therefore, the bidder could miss out on potentially valuable items. Moreover, if the platform wanted to guarantee that the bidder does not spend more than the budget $B$, it would be enough to set a *virtual budget* of $B - \tilde{O}(T^{1/2})$ to compensate for the potential constraints violation. Finally, we argue that, in large-scale markets, an individual bidder has almost no impact on prices, and, thus, stochastic behavior of costs is a reasonable assumption.

**Adversarial case.**   Theorem 6.1 of [17] shows how to construct an algorithm that provides a $\rho$ fraction of the optimal utility for problems with budget constraints and adversarial inputs. The ratio $\rho/(1 + \rho)$ obtained in Theorem 6.1 matches such result. The latter assumes that rewards and costs are in $[0, 1]$, and, thus, $g_t \in [-\rho, 1 - \rho]$ (as they only model budget constraints). However, in our case we have $g_t \in [-1, 1]$. By normalizing the former range to match with ours, we get $g_t \in [-\rho/(1-\rho), 1]$. Therefore, the feasibility parameter would be $\rho' = \rho/(1 - \rho)$. By rewriting our guarantees as a function of $\rho$, we get $\rho'/(1 + \rho') = \rho$, which is the same guarantee of [17].

**Handling ROI constraints.**   Traditional budget-pacing mechanisms (see, *e.g.*, [11, 8]) are based on primal-dual algorithms that are near optimal in settings with knapsack constraints only, and they cannot be generalized to deal with other types of long-term constraints. However, there are many real-world situations in which guaranteeing other types of constraints is crucial for practical applications (see, *e.g.*, [26, 25]). One example is the case of *return on investment* (ROI) constraints [6, 26, 32]. The recent work by Golrezaei et al. [25] presents a threshold-based algorithm for repeated second-price auctions under budget and ROI constraints. Our framework allows advertisers to reach a target ROI while keeping expenses under control also in the setting of repeated first-price auctions. In particular, given a target ROI $\omega$, we define the ROI constraints as $g_t(b_t) = (\omega - v_t/b_t)\mathbb{1}\{b_t \geq \beta_t\} \leq 0$. Then, it is enough to instantiate our framework as described before to immediately get that the cumulative violation of the budget and ROI constraints are upper bounded by $\tilde{O}(T^{1/2})$. This holds both in the fully stochastic and in the fully adversarial setting.

See Appendix C for a detailed discussion on the types of constraints that our framework can handle.

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
