- 1$ | 5:   $\mathcal{R}^{\text{P}}.\text{OBSERVEUTILITY}(u_t^{\text{P}})$     ▷ Full f. |
|     ▷ Phase II: Recovery |     $\mathcal{R}^{\text{P}}.\text{OBSERVEUTILITY}(u_t^{\text{P}}(\boldsymbol{x}_t))$   ▷ Bandit f. |
| 8: $\mathcal{R}^{\text{P}}_{\text{II}} \leftarrow \text{INIT}^{\text{P}}\left(\mathcal{X}, [-1, 1], \eta\right)$ |     ▷ Dual RM update |
| 9: $\mathcal{R}^{\text{D}}_{\text{II}} \leftarrow \text{INIT}^{\text{D}}\left(\Delta_m, [-1, 1], 0\right)$ | 6: Let $u_t^{\text{D}} : \boldsymbol{\lambda} \mapsto \langle \boldsymbol{\lambda}, g_t(\boldsymbol{x}) \rangle$ |
| 10: **while** $t \leq T$ **do** | 7: $\mathcal{R}^{\text{D}}.\text{OBSERVEUTILITY}(u_t^{\text{D}})$ |
| 11:     $\boldsymbol{x}_t \leftarrow \text{LAGRANGIANGAME}(\mathcal{R}^{\text{P}}_{\text{II}}, \mathcal{R}^{\text{D}}_{\text{II}}, 0)$ | |
| 12:     $t \leftarrow t + 1$ | |

to build RMs working in the primal domain and the latter for those operating on the dual one. Their actual implementation depends on the specific problem at hand. In the following, we let $\mathcal{E}^{\text{P}}_{t,\eta}$ be the regret upper bound (on $t \in [T]$ rounds) for primal RMs $\mathcal{R}^{\text{P}}$ dealing with utility functions having range $[0, 1]$, as returned by the call $\text{INIT}^{\text{P}}(\mathcal{X}, [0, 1], \eta)$. Notice that, when the range is $[a, b]$, the same RM can be adopted by first normalizing utility values, so that the resulting regret upper bound is $(b - a)\mathcal{E}^{\text{P}}_{t,\eta}$. As for dual RMs $\mathcal{R}^{\text{D}}$, we let $\mathcal{E}^{\text{D}}_t$ be the regret upper bound (on $t \in [T]$ rounds) provided by the RM defined for the set $\Delta_m$, while $\mathcal{E}^{\text{D}}_t / \tilde{\rho}$ is the upper bound for the dual RM instantiated on the set $\mathcal{D}_{\tilde{\rho}}$. Notice that, since dual RMs always have full feedback, we can safely assume that the regret bounds $\mathcal{E}^{\text{D}}_t$ hold deterministically. We also assume that RMs provide bounds that increase with the number of rounds, *i.e.*, such that $\mathcal{E}^{\text{P}}_{t,\eta} \leq \mathcal{E}^{\text{P}}_{t',\eta}$ and $\mathcal{E}^{\text{D}}_t \leq \mathcal{E}^{\text{D}}_{t'}$

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

# A Related works

The *online convex optimization* (OCO) framework was first proposed in the machine learning literature by Zinkevich [46], and since then it has significantly expanded becoming widely influential in the learning community (see, *e.g.*, [27, 29, 38]). In what follows, we highlight the most relevant works with respect to ours from the literature related to online convex optimization problems with constraints. The analysis and the results are quite different depending on the nature of the constraints, which may be static, *i.e.*, time-invariant, or stochastic/adversarial, *i.e.*, time-variant.

**Static constraints.** Zinkevich [46] first addressed online convex optimization problems with static constraints by developing a projection-based *online gradient descent* (OGD) algorithm. This method guarantees a regret upper bound of $\mathcal{O}(\sqrt{T})$ for an arbitrary sequence of convex objective functions with bounded subgradients. Hazan et al. [28] showed that this is a tight bound up to constant factors. When the set defined by the static constraints is complex, the conventional projection-based online algorithms can be difficult to implement due to the potentially high computational cost of carrying out the projection operation. To overcome this difficulty, Mahdavi et al. [35] propose an efficient algorithm which is an adaptation of OGD achieving a cumulative regret of order $\mathcal{O}(\sqrt{T})$ and a cumulative constraints violation of $\mathcal{O}(T^{3/4})$. These bounds are generalized by Jenatton et al. [31] who propose an algorithm that achieves a cumulative regret of $\mathcal{O}(T^{\max\{\beta,1-\beta\}})$ and a cumulative violation of $\mathcal{O}(T^{1-\beta/2})$, where $\beta \in (0,1)$ is a user-defined parameter. Other works, such as, *e.g*, [45, 43], propose primal-dual algorithms and achieve better bounds by making further assumptions. In particular, Yu and Neely [43] achieve bounds on the cumulative regret of $\mathcal{O}(\sqrt{T})$ and on the cumulative violation of $\mathcal{O}(1)$ by assuming that the Slater's condition holds (*i.e.*, the existence of a strictly feasible solution). Then, Yuan and Lamperski [45] achieve a cumulative regret of $\mathcal{O}(\log T)$ and a constraint violation of $\mathcal{O}(\sqrt{T})$ under the assumption that the objective functions are strongly convex. In all the the papers cited above, the regret is computed with respect to the best fixed action in hindsight,that does *not* violate the constraints at each round $t$. This metric is called *static regret*.

**Stochastic constraints.** Yu et al. [44] consider an online convex optimization framework with stochastic constraints, where the objective functions are chosen by an adversary, and the constraint functions are independent and identically distributed (i.i.d.) over time. Yu et al. [44] provide a primal-dual proximal gradient algorithm achieving $\mathcal{O}(\sqrt{T})$ cumulative regret and constraint violation by assuming Slater's condition. Moreover, Wei et al. [41] provide bounds of the same order by assuming a less stringent version of the Slater's condition. As a performance metric, the latter work use *static regret*.

**Adversarial constraints.** Various works in the literature address the online learning setting with adversarial reward and constraint functions. This problem was first studied by Mannor et al. [36] in a two-player game setting. The regret is computed with respect to the best strategy from the set of fixed strategies that satisfy the constraints on average. Mannor et al. [36] show that in general it is impossible to compete against the best decision in such a set. In particular, they construct a two-player game where there exists a policy for the adversary such that, among the policies of the player that violate sublinearly the constraints, there is no policy that can achieve the no-regret property in terms of maximizing the player's reward. Sun et al. [40] study a similar problem related to contextual bandits and show that also in their setting the decision maker is unable to compete again this baseline by adapting the result from Mannor et al. [36] to their setting. To circumvent this issue and provide some guarantees, they rely on a weaker baseline to compute the regret. In particular, they assume that the decision set is rich enough that, in hindsight, there exist a fixed action that satisfies the constraints at each round: they are using the *static regret* as a performance metric. Then, by employing static regret as a baseline, Sun et al. [40] show that the approaches of Mahdavi et al. [35] and Jenatton et al. [31] can be extended to the online learning framework with adversarial sequential constraints. Therefore, they provide an algorithm which is a generalization of that from Mahdavi et al. [35] achieving sublinear cumulative regret and constraint violations.

Liakopoulos et al. [33] define a new notion of regret, to overcome the impossibility result from Mannor et al. [36]. They introduce a refined regret metric which compares the agent's incurred losses to those of a *K-benchmark*, which is the best strategy in the hindsight such that, for each time window of length $K$, the long-term constraints over that window are satisfied. They provide parametric results

that are useful to balance the trade-off between regret minimization and long-term residual constraint violation. Moreover, instead of the Slater's condition they consider a less stringent assumption related to the definition of their regret metric.

A recent line of works such as [21, 20] and [16] provide some results related to the regret against *dynamic policies*. As expected, comparing against a dynamic baseline require very strong assumptions. Chen et al. [21] compute a bound on the cumulative dynamic regret which is sublinear in the time horizon $T$ only if the drift of the baseline sequence (*i.e.*, $\sum_{t=1}^{T} ||\boldsymbol{x}_{t+1}^* - \boldsymbol{x}_t^*||$) and that of the constraints (*i.e.*, $\sum_{t=1}^{T} \max_{\boldsymbol{x}} ||[\boldsymbol{g}_{t+1}(\boldsymbol{x}) - \boldsymbol{g}_t(\boldsymbol{x})]^+||$) are $o(T^{2/3})$. Cao and Liu [16] consider a bandit feedback setting and, in order to provide sublinear regret and constraint violations, they assume that all the loss functions have uniformly bounded difference (*i.e.*, for each $t$ and $\boldsymbol{x}, \boldsymbol{x}' \in \mathcal{X}$, $|f_t(\boldsymbol{x}) - f_t(\boldsymbol{x}')| \leq M$ for some positive constant $M$), and that the drift of the baseline sequence is sublinear. In other words, the underlying dynamic optimization problems vary *slowly* over time. Both Chen et al. [21] and Cao and Liu [16] need to assume the Slater's condition. Yi et al. [42] provide similar results in a distributed online convex optimization setting with adversarial constraints. They analyze both the case in which the Slater's condition holds, and the case without this assumption.

Others relevant related works, are those studying online learning problems in which the decision maker has to satisfy supply/budget constraints. In this setting, the decision maker wants to maximize their expected reward without violating a set of $m$ resource constraints. The process stops at time horizon $T$, or when the total consumption of some resource exceeds its budget. Badanidiyuru et al. [7] first introduce and solve the *Bandits with Knapsacks* (BwK) framework, in which thay consider bandit feedback, stochastic objective and constraint functions. Other optimal algorithms for Stochastic BwK were proposed by Agrawal and Devanur [2, 3] and by Immorlica et al. [30]. The *Adversarial Bandits with Knapsacks* setting was first studied by Immorlica et al. [30]. The authors shows that an appropriate baseline is the best fixed distribution over arms. Achieving no-regret is no longer possible under this baseline and, therefore, they provide no-$\alpha$-regret guarantees for their algorithm.

We remark that in this paper we are able to handle more general constraints than Immorlica et al. [30], which can deal only with budget constraints. Moreover, we can compete with a baseline stronger than the static regret used by Sun et al. [40], without needing the strong assumptions required, for instance, by Cao and Liu [16].

## B  Omitted proofs

### B.1  Proof omitted from Section 2.1

**Theorem 2.3.** *Given $f \in \mathcal{F}$ and $g \in \mathcal{G}$ such that $d_g > 0$, it holds*

$$\sup_{\boldsymbol{\xi} \in \Xi} \inf_{\boldsymbol{\lambda} \in \mathcal{D}_{d_g}} \mathcal{L}_{f,g}(\boldsymbol{\xi}, \boldsymbol{\lambda}) = \inf_{\boldsymbol{\lambda} \in \mathcal{D}_{d_g}} \sup_{\boldsymbol{\xi} \in \Xi} \mathcal{L}_{f,g}(\boldsymbol{\xi}, \boldsymbol{\lambda}) = \text{OPT}_{f,g}.$$

*Proof.* As a first step, we prove that $\inf_{\boldsymbol{\lambda} \in \mathcal{D}_{d_g}} \sup_{\boldsymbol{\xi} \in \Xi} \mathcal{L}(\boldsymbol{\xi}, \boldsymbol{\lambda}) = \inf_{\boldsymbol{\lambda} \in \mathbb{R}_+^m} \sup_{\boldsymbol{\xi} \in \Xi} \mathcal{L}(\boldsymbol{\xi}, \boldsymbol{\lambda})$. Notice that for each $\boldsymbol{\lambda} \in \mathbb{R}_+^m$ such that $\|\boldsymbol{\lambda}\|_1 > 1/d_g$, it holds

$$\sup_{\boldsymbol{\xi} \in \Xi} \mathcal{L}(\boldsymbol{\xi}, \boldsymbol{\lambda}) \geq \mathcal{L}(\boldsymbol{\xi}^\circ, \boldsymbol{\lambda}) \geq -\langle \boldsymbol{\lambda}^*, g(\boldsymbol{\xi}^\circ) \rangle \geq d_g \|\boldsymbol{\lambda}^*\|_1 > 1,$$

where, with an abuse of notation, $\boldsymbol{\xi}^\circ \in \Xi$ denotes a strictly feasible strategy mixture for Problem $\text{LP}_{f,g}$. That is a strategy mixture $\boldsymbol{\xi} \in \Xi$ which is optimal for the problem defining $d_g$ in Equation (1), and, thus, it satisfies all the constraints by at least $d_g$ (*i.e.*, it holds $g_i(\boldsymbol{\xi}^\circ) \leq -d_g$ for all $i \in [m]$).[8] Thus, it holds that

$$\inf_{\boldsymbol{\lambda} \in \mathbb{R}_+^m \setminus \mathcal{D}_{d_g}} \sup_{\boldsymbol{\xi} \in \Xi} \mathcal{L}(\boldsymbol{\xi}, \boldsymbol{\lambda}) > 1.$$

Moreover, since

$$\inf_{\boldsymbol{\lambda} \in \mathcal{D}_{d_g}} \sup_{\boldsymbol{\xi} \in \Xi} \mathcal{L}(\boldsymbol{\xi}, \boldsymbol{\lambda}) \leq \sup_{\boldsymbol{\xi} \in \Xi} \mathcal{L}(\boldsymbol{\xi}, \boldsymbol{0}) \leq 1,$$

---

[8]Notice that $\boldsymbol{\xi}^\circ$ may *not* be well defined when the problem in Equation 1 does *not* admit a maximum. In such cases, we can take a $\boldsymbol{\xi}^\circ$ that is arbitrarily "close" to a supremum, so that the result still holds.

we can conclude that

$$\inf_{\boldsymbol{\lambda}\in\mathbb{R}_+^m}\sup_{\boldsymbol{\xi}\in\Xi}\mathcal{L}(\boldsymbol{\xi},\boldsymbol{\lambda})=\min\left\{\inf_{\boldsymbol{\lambda}\in\mathcal{D}_{d_g}}\sup_{\boldsymbol{\xi}\in\Xi}\mathcal{L}(\boldsymbol{\xi},\boldsymbol{\lambda});\inf_{\boldsymbol{\lambda}\in\mathbb{R}_+^m\setminus\mathcal{D}_{d_g}}\sup_{\boldsymbol{\xi}\in\Xi}\mathcal{L}(\boldsymbol{\xi},\boldsymbol{\lambda})\right\}$$

$$=\inf_{\boldsymbol{\lambda}\in\mathcal{D}_{d_g}}\sup_{\boldsymbol{\xi}\in\Xi}\mathcal{L}(\boldsymbol{\xi},\boldsymbol{\lambda}). \tag{3}$$

Then,

$$\begin{aligned}
\mathtt{OPT}_{f,g}&=\sup_{\boldsymbol{\xi}\in\Xi}\inf_{\boldsymbol{\lambda}\in\mathbb{R}_+^m}\mathcal{L}(\boldsymbol{\xi},\boldsymbol{\lambda})\\
&\leq\sup_{\boldsymbol{\xi}\in\Xi}\inf_{\boldsymbol{\lambda}\in\mathcal{D}_{d_g}}\mathcal{L}(\boldsymbol{\xi},\boldsymbol{\lambda})\\
&\leq\inf_{\boldsymbol{\lambda}\in\mathcal{D}_{d_g}}\sup_{\boldsymbol{\xi}\in\Xi}\mathcal{L}(\boldsymbol{\xi},\boldsymbol{\lambda})\\
&=\inf_{\boldsymbol{\lambda}\in\mathbb{R}_+^m}\sup_{\boldsymbol{\xi}\in\Xi}\mathcal{L}(\boldsymbol{\xi},\boldsymbol{\lambda})\\
&=\mathtt{OPT}_{f,g},
\end{aligned}$$

where the first inequality holds since in the right-hand side the inf is taken over the more restrictive set $\mathcal{D}_{d_g}$, the second one by the *max–min inequality*, while the second-to-last equality holds by Equation (3). This concludes the proof. $\qquad\square$

## B.2 Proofs omitted from Section 4

**Lemma 4.2.** *After running Algorithm 1, the event $\mathbf{E}$ holds with probability at least $1-\delta$.*

*Proof.* Given a desired failure probability $\delta\in(0,1)$, recall that $\eta=\delta/3$ and set $\varepsilon=\eta/18mT^2$. Consider the following inequalities in which the differences between expectations and empirical means of constraint functions are bounded:

$$\left|\sum_{\tau=t}^{t'}g_{\tau,i}(\boldsymbol{x}_\tau)-\sum_{\tau=t}^{t'}\bar{g}_i(\boldsymbol{x}_\tau)\right|>2\sqrt{2(t'-t)\ln\frac{1}{\varepsilon}}, \tag{4}$$

$$\left|\sum_{\tau=t}^{t'}g_{\tau,i}(\boldsymbol{\xi}^\circ)-\sum_{\tau=t}^{t'}\bar{g}_i(\boldsymbol{\xi}^\circ)\right|>2\sqrt{2(t'-t)\ln\frac{1}{\varepsilon}}, \tag{5}$$

$$\left|\sum_{\tau=t}^{t'}g_{\tau,i}(\boldsymbol{\xi}^*)-\sum_{\tau=t}^{t'}\bar{g}_i(\boldsymbol{\xi}^*)\right|>2\sqrt{2(t'-t)\ln\frac{1}{\varepsilon}}, \tag{6}$$

$$\left|\sum_{\tau=t}^{t'}\boldsymbol{\lambda}_\tau g_{\tau,i}(\boldsymbol{x}_\tau)-\sum_{\tau=t}^{t'}\boldsymbol{\lambda}_\tau\bar{g}_i(\boldsymbol{x}_\tau)\right|>2\max_{\tau\in[T]:t\leq\tau\leq t'}||\boldsymbol{\lambda}_\tau||_1\sqrt{2(t'-t)\ln\frac{1}{\varepsilon}}, \tag{7}$$

$$\left|\sum_{\tau=t}^{t'}\boldsymbol{\lambda}_\tau\,g_{\tau,i}(\boldsymbol{\xi}^*)-\sum_{\tau=t}^{t'}\boldsymbol{\lambda}_\tau\,\bar{g}_i(\boldsymbol{\xi}^*)\right|>2\max_{\tau\in[T]:t\leq\tau\leq t'}||\boldsymbol{\lambda}_\tau||_1\sqrt{2(t'-t)\ln\frac{1}{\varepsilon}}, \tag{8}$$

$$\left|\sum_{\tau=t}^{t'}\boldsymbol{\lambda}_\tau\,g_{\tau,i}(\boldsymbol{\xi}^\circ)-\sum_{\tau=t}^{t'}\boldsymbol{\lambda}_\tau\,\bar{g}_i(\boldsymbol{\xi}^\circ)\right|\leq2\max_{\tau\in[T]:t\leq\tau\leq t'}||\boldsymbol{\lambda}_\tau||_1\sqrt{2(t'-t)\ln\frac{1}{\varepsilon}}. \tag{9}$$

By applying Azuma-Hoeffding inequality to each martingale difference sequence, we get that each inequality holds with probability at most $2\varepsilon$. We denote by $\mathbf{E}_\eta$ the event in which Equations (4), (5), (6), and (7) are satisfied for all $t,t'\in[T]$ with $t<t'$, and for all $i\in[m]$. By a union bound that takes into account the six events above, the $m$ constraints, and all the possible time intervals from $t$ to $t'$, which are at most $T^2$, we get:

$$\mathbb{P}(\mathbf{E}_\eta)\geq1-6mT^2(2\varepsilon)=1-12mT^2\varepsilon=1-\frac{2}{3}\eta\geq1-\eta.$$

Therefore, event $\mathbf{E}_\eta$ holds with probability at least $1 - \eta$. Moreover, let us recall that:

$$\mathcal{E}_{t'-t,\eta} = \sqrt{8(t'-t)\ln\frac{18mT^2}{\eta}} = 2\sqrt{2(t'-t)\ln\frac{1}{\varepsilon}}.$$

Now, consider event $\mathbf{E}$ in which Algorithms 1 satisfies the following conditions: (i) the regret incurred by $\mathcal{R}_I^P$ after $T_1$ rounds is upper bounded by $\mathcal{E}_{T_1,\eta}^P$; (ii) the regret cumulated by $\mathcal{R}_{II}^P$ after the remaining $T - T_1$ rounds is upper bounded by $\mathcal{E}_{T-T_1,\eta}^P$; and (iii) event $\mathbf{E}_\eta$ holds. Recall that each one of the conditions (i), (ii) and (iii) holds with probability at least $1 - \eta$; hence, by a union bound we get:

$$\mathbb{P}(\mathbf{E}) \geq 1 - 3\eta = 1 - \delta.$$

This concludes the proof. $\qquad\square$

**Lemma 4.3.** *If event $\mathbf{E}$ holds, then after round $T_1$ of Algorithm 1 the following inequality holds:*
$\sum_{t=1}^{T_1} f_t(\boldsymbol{x}_t) \geq \sum_{t=1}^{T_1} f_t(\boldsymbol{\xi}^*) + (T - T_1) - \frac{1}{\tilde{\rho}}\mathcal{E}_{T_1,\eta} - \left(1 + \frac{2}{\tilde{\rho}}\right)\mathcal{E}_{T_1,\eta}^P - \frac{1}{\tilde{\rho}}\mathcal{E}_{T_1}^D.$

*Proof.* By the no-regret property of the primal regret minimizer, we have that:

$$\sum_{t=1}^{T_1}\left(f_t(\boldsymbol{x}_t) - \langle \boldsymbol{\lambda}_t, g_t(\boldsymbol{x}_t)\rangle\right) \geq \sum_{t=1}^{T_1}\left(f_t(\boldsymbol{\xi}^*) - \langle \boldsymbol{\lambda}_t, g_t(\boldsymbol{\xi}^*)\rangle\right) - \left(1 + \frac{2}{\tilde{\rho}}\right)\mathcal{E}_{T_1,\eta}^P. \qquad (10)$$

Let $i^\star \in \arg\max_{i\in[m]} \sum_{t=1}^{T_1} g_{t,i}(\boldsymbol{x}_t)$ be one of the "most violated" constraints. We prove that:

$$\sum_{t=1}^{T_1}\langle \boldsymbol{\lambda}_t, g_t(\boldsymbol{x}_t)\rangle \geq (T - T_1) - \frac{1}{\tilde{\rho}}\mathcal{E}_{T_1}^D. \qquad (11)$$

To do that, we consider the following two cases.

**Case $T_1 = T$.** We get:

$$\sum_{t=1}^{T_1}\langle \boldsymbol{\lambda}_t, g_t(\boldsymbol{x}_t)\rangle \geq \sum_{t=1}^{T_1}\langle \boldsymbol{0}, g_t(\boldsymbol{x}_t)\rangle - \frac{1}{\tilde{\rho}}\mathcal{E}_{T_1}^D = (T - T_1) - \frac{1}{\tilde{\rho}}\mathcal{E}_{T_1}^D.$$

**Case $T_1 < T$.** By the condition in Line 4 of Algorithm 1, we have that $\sum_{t=1}^{T_1} g_{t,i^\star}(\boldsymbol{x}_t) \geq (T - T_1)\tilde{\rho}$. Thus, we have that:

$$\sum_{t=1}^{T_1}\langle \boldsymbol{\lambda}_t, g_t(\boldsymbol{x}_t)\rangle \geq \sum_{t=1}^{T_1} \frac{1}{\tilde{\rho}} g_{t,i^\star}(\boldsymbol{x}_t) - \frac{1}{\tilde{\rho}}\mathcal{E}_{T_1}^D \geq (T - T_1) - \frac{1}{\tilde{\rho}}\mathcal{E}_{T_1}^D,$$

where the first inequality follows from the no-regret property of the dual regret minimizer and the second one from the fact that $\sum_{t=1}^{T_1} g_{t,i^\star}(\boldsymbol{x}_t) \geq (T - T_1)\tilde{\rho}$ when $T_1 < T$.

Now, by using Equation (11), we can provide a lower bound on the cumulative reward obtained by Algorithm 1 during the play phase. We have that:

$$\sum_{t=1}^{T_1} f_t(\boldsymbol{x}_t) \geq \sum_{t=1}^{T_1}\left(f_t(\boldsymbol{\xi}^*) - \langle \boldsymbol{\lambda}_t, g_t(\boldsymbol{\xi}^*)\rangle + \langle \boldsymbol{\lambda}_t, g_t(\boldsymbol{x}_t)\rangle\right) - \left(1 + \frac{2}{\tilde{\rho}}\right)\mathcal{E}_{T_1,\eta}^P,$$

$$\geq \sum_{t=1}^{T_1}\left(f_t(\boldsymbol{\xi}^*) - \langle \boldsymbol{\lambda}_t, g_t(\boldsymbol{\xi}^*)\rangle\right) + (T - T_1) - \left(1 + \frac{2}{\tilde{\rho}}\right)\mathcal{E}_{T_1,\eta}^P - \frac{1}{\tilde{\rho}}\mathcal{E}_{T_1}^D,$$

$$\geq \sum_{t=1}^{T_1}\left(f_t(\boldsymbol{\xi}^*) - \langle \boldsymbol{\lambda}_t, \bar{g}(\boldsymbol{\xi}^*)\rangle\right) + (T - T_1) - \left(1 + \frac{2}{\tilde{\rho}}\right)\mathcal{E}_{T_1,\eta}^P - \frac{1}{\tilde{\rho}}\mathcal{E}_{T_1}^D - \frac{1}{\tilde{\rho}}\mathcal{E}_{T_1,\eta},$$

$$\geq \sum_{t=1}^{T_1} f_t(\boldsymbol{\xi}^*) + (T - T_1) - \left(1 + \frac{2}{\tilde{\rho}}\right)\mathcal{E}_{T_1,\eta}^P - \frac{1}{\tilde{\rho}}\mathcal{E}_{T_1}^D - \frac{1}{\tilde{\rho}}\mathcal{E}_{T_1,\eta},$$

where the first inequality holds by Equation (10), the second one by Equation (11), the third one follows from the fact that the event $\mathbf{E}$ holds, while the last one from the fact that $\bar{g}(\boldsymbol{\xi}^*) \leq 0$ by definition. $\qquad\square$

**Lemma 4.4.** *If event* **E** *holds, then after Algorithm 1 halts, the following holds for every* $i \in [m]$:
$\sum_{t=T_1+1}^{T} g_{t,i}(\boldsymbol{x}_t) \leq -(T - T_1)\rho + 2\mathcal{E}^P_{T-T_1,\eta} + \mathcal{E}^D_{T-T_1} + \mathcal{E}_{T-T_1,\eta}.$

*Proof.* Let $i^\star \in \arg\max_{i \in [m]} \sum_{t=T_1+1}^{T} g_{t,i}(\boldsymbol{x}_t)$ be one of the "most violated" constraints. Then,

$$
\begin{aligned}
(T - T_1)\rho &\leq - \sum_{t=T_1+1}^{T} \langle \boldsymbol{\lambda}_t, \bar{g}(\boldsymbol{\xi}^\circ) \rangle \\
&\leq - \sum_{t=T_1+1}^{T} \langle \boldsymbol{\lambda}_t g_t(\boldsymbol{\xi}^\circ) \rangle + \mathcal{E}_{T-T_1,\eta} \\
&\leq - \sum_{t=T_1+1}^{T} \langle \boldsymbol{\lambda}_t g_t(\boldsymbol{x}_t) \rangle + 2\mathcal{E}^P_{T-T_1,\eta} + \mathcal{E}_{T-T_1,\eta} \\
&\leq - \sum_{t=T_1+1}^{T} g_{t,i^\star}(\boldsymbol{x}_t) + \mathcal{E}^D_{T-T_1} + 2\mathcal{E}^P_{T-T_1,\eta} + \mathcal{E}_{T-T_1,\eta},
\end{aligned}
$$

where the first inequality comes from the definition of $\rho$, the second one from the fact that event **E** holds, the third one from the no-regret property of the primal regret minimizer, and the last one from the no-regret property of the dual regret minimizer. Hence,

$$
\sum_{t=T_1+1}^{T} g_{t,i^\star}(\boldsymbol{x}_t) \leq -(T - T_1)\rho - \mathcal{E}^D_{T-T_1} + 2\mathcal{E}^P_{T-T_1,\eta} + \mathcal{E}_{T-T_1,\eta}. \tag{12}
$$

It follows from the definition of $i^\star$ that, if Equation (12) holds for $i^\star$, then, it holds for every $i \in [m]$. This concludes the proof. $\square$

**Theorem 4.6.** *Suppose that functions* $f_t$ *and* $g_t$ *are selected adversarially and stochastically, respectively. If Condition 4.5 is satisfied, then, with probability at least* $1 - \delta$, *Algorithm 1 provides* $R^T \leq \frac{1}{\tilde{\rho}}\mathcal{E}_{T,\eta} + \left(1 + \frac{2}{\tilde{\rho}}\right)\mathcal{E}^P_{T,\eta} + \frac{1}{\tilde{\rho}}\mathcal{E}^D_T$ *and* $V^T \leq M_{\tilde{\rho}} + 2\mathcal{E}^P_{T,\eta} + \mathcal{E}^D_T + \mathcal{E}_{T,\eta}.$

*Proof.* By Lemma 4.2, event **E** holds with probability at least $1 - \delta$. In the rest of the proof, we assume the event **E** holds, providing a bound that holds with probability at least $1 - \delta$.

We first provide an upper bound on the cumulative regret. By Lemma 4.3, we have:

$$
\sum_{t=1}^{T_1} f_t(\boldsymbol{x}_t) \geq \sum_{t=1}^{T_1} f_t(\boldsymbol{\xi}^*) + (T - T_1) - \frac{1}{\tilde{\rho}}\mathcal{E}_{T_1,\eta} - \left(1 + \frac{2}{\tilde{\rho}}\right)\mathcal{E}^P_{T_1,\eta} - \frac{1}{\tilde{\rho}}\mathcal{E}^D_{T_1}. \tag{13}
$$

Hence, it holds:

$$
\begin{aligned}
\sum_{t=1}^{T} f_t(\boldsymbol{x}_t) &\geq \sum_{t=1}^{T_1} f_t(\boldsymbol{x}_t) \\
&\geq \sum_{t=1}^{T_1} f_t(\boldsymbol{\xi}^*) + (T - T_1) - \frac{1}{\tilde{\rho}}\mathcal{E}_{T_1,\eta} - \left(1 + \frac{2}{\tilde{\rho}}\right)\mathcal{E}^P_{T_1,\eta} - \frac{1}{\tilde{\rho}}\mathcal{E}^D_{T_1} \\
&\geq \sum_{t=1}^{T} f_t(\boldsymbol{\xi}^*) - \frac{1}{\tilde{\rho}}\mathcal{E}_{T_1,\eta} - \left(1 + \frac{2}{\tilde{\rho}}\right)\mathcal{E}^P_{T_1,\eta} - \frac{1}{\tilde{\rho}}\mathcal{E}^D_{T_1} \\
&\geq \sum_{t=1}^{T} f_t(\boldsymbol{\xi}^*) - \frac{1}{\tilde{\rho}}\mathcal{E}_{T,\eta} - \left(1 + \frac{2}{\tilde{\rho}}\right)\mathcal{E}^P_{T,\eta} - \frac{1}{\tilde{\rho}}\mathcal{E}^D_T,
\end{aligned}
$$

where the second inequality holds by Equation (13) and the third one by $\sum_{t=T_1+1}^{T} f_t(\boldsymbol{\xi}^*) \leq T - T_1$, which follows from the fact that the range of $f_t$ is $[0, 1]$.

By recalling that $\boldsymbol{\xi}^* \in \Xi$ is defined as an optimal solution to Problem $\mathrm{LP}_{\bar{f},\bar{g}}$ and $R^T = T\,\mathrm{OPT}_{\bar{f},\bar{g}} - \sum_{t=1}^T f_t(\boldsymbol{x}_t)$, the following bound on the cumulative regret holds:

$$R^T = \sum_{t=1}^T f_t(\boldsymbol{\xi}^*) - \sum_{t=1}^T f_t(\boldsymbol{x}_t) \le \frac{1}{\rho}\mathcal{E}_{T,\eta} + \left(1 + \frac{2}{\rho}\right)\mathcal{E}_{T,\eta}^{\mathrm{P}} + \frac{1}{\rho}\mathcal{E}_T^{\mathrm{D}}.$$

Next, we provide an upper bound on the cumulative constraints violation.

By Lemma 4.4, for every $i \in [m]$, we have that:

$$\sum_{t=T_1+1}^T g_{t,i}(\boldsymbol{x}_t) \le -(T - T_1)\rho + 2\mathcal{E}_{T-T_1,\eta}^{\mathrm{P}} + \mathcal{E}_{T-T_1}^{\mathrm{D}} + \mathcal{E}_{T-T_1,\eta}. \tag{14}$$

Hence, for every $i \in [m]$, it holds

$$
\begin{aligned}
\sum_{t=1}^T g_{t,i}(\boldsymbol{x}_t) &= \sum_{t=1}^{T_1} g_{t,i}(\boldsymbol{x}_t) + \sum_{t=T_1+1}^T g_{t,i}(\boldsymbol{x}_t) \\
&\le (T - T_1)\tilde{\rho} + M_{\tilde{\rho}} - (T - T_1)\rho + 2\mathcal{E}_{T-T_1,\eta}^{\mathrm{P}} + \mathcal{E}_{T-T_1}^{\mathrm{D}} + \mathcal{E}_{T-T_1,\eta} \\
&\le M_{\tilde{\rho}} + 2\mathcal{E}_{T-T_1,\eta}^{\mathrm{P}} + \mathcal{E}_{T-T_1}^{\mathrm{D}} + \mathcal{E}_{T-T_1,\eta} \\
&\le M_{\tilde{\rho}} + 2\mathcal{E}_{T,\eta}^{\mathrm{P}} + \mathcal{E}_T^{\mathrm{D}} + \mathcal{E}_{T,\eta}.
\end{aligned}
$$

The first inequality follows from Equation (14) and by the condition in Line 4 of Algorithm 1, which ensures $\sum_{t=1}^{T_1} g_{t,i}(\boldsymbol{x}_t) \le (T - T_1)\tilde{\rho} + M_{\tilde{\rho}}$ for every $i \in [m]$. Moreover, the second inequality follows from $\tilde{\rho} \le \rho$, since Condition 4.5 holds. Let $i^\star \in \arg\max_{i \in [m]} \sum_{t=1}^T g_{t,i}(\boldsymbol{x}_t)$ be one of the most violated constraints. By recalling that $V^T = \max_{i \in [m]} \sum_{t=1}^T g_{t,i}(\boldsymbol{x}_t)$, the following bound on the cumulative constraints violation holds:

$$V^T = \sum_{t=1}^T g_{t,i^\star}(\boldsymbol{x}_t) \le M_{\tilde{\rho}} + 2\mathcal{E}_{T,\eta}^{\mathrm{P}} + \mathcal{E}_T^{\mathrm{D}} + \mathcal{E}_{T,\eta}.$$

This concludes the proof. $\qquad\square$

**Theorem 4.7.** *Suppose that functions $f_t$ and $g_t$ are selected adversarially and stochastically, respectively. Algorithm 1 guarantees that the following bounds hold with probability at least $1 - \delta$:*
$R_T \le T^{1/4}\mathcal{E}_{T,\eta} + \left(1 + 2T^{1/4}\right)\mathcal{E}_{T,\eta}^{\mathrm{P}} + T^{1/4}\mathcal{E}_T^{\mathrm{D}}$ *and* $V_T \le T^{3/4} + M_{T^{-1/4}} + 2\mathcal{E}_{T,\eta}^{\mathrm{P}} + \mathcal{E}_T^{\mathrm{D}} + \mathcal{E}_{T,\eta}.$

*Proof.* If $\hat{\rho} \ge 2T^{-1/4}$, the claim follows by Theorem 4.6. Thus, we prove the statement for the case $\tilde{\rho} = T^{-1/4}$. First, we provide an upper bound on the cumulative regret. By Lemma 4.2, we have that event the $\mathbf{E}$ holds with probability at least $1 - \delta$. In the rest of the proof, we assume that the event $\mathbf{E}$ holds, and provide a bound that holds with probability at least $1 - \delta$. We have:

$$
\begin{aligned}
\sum_{t=1}^T f_t(\boldsymbol{x}_t) &\ge \sum_{t=1}^{T_1} f_t(\boldsymbol{x}_t) \\
&\ge \sum_{t=1}^{T_1} f_t(\boldsymbol{\xi}^*) + (T - T_1) - \frac{1}{\tilde{\rho}}\mathcal{E}_{T_1,\eta} - \left(1 + \frac{2}{\tilde{\rho}}\right)\mathcal{E}_{T_1,\eta}^{\mathrm{P}} - \frac{1}{\tilde{\rho}}\mathcal{E}_{T_1}^{\mathrm{D}} \\
&\ge \sum_{t=1}^T f_t(\boldsymbol{\xi}^*) - \frac{1}{\tilde{\rho}}\mathcal{E}_{T_1,\eta} - \left(1 + \frac{2}{\tilde{\rho}}\right)\mathcal{E}_{T_1,\eta}^{\mathrm{P}} - \frac{1}{\tilde{\rho}}\mathcal{E}_{T_1}^{\mathrm{D}} \\
&\ge \sum_{t=1}^T f_t(\boldsymbol{\xi}^*) - \frac{1}{\tilde{\rho}}\mathcal{E}_{T,\eta} - \left(1 + \frac{2}{\tilde{\rho}}\right)\mathcal{E}_{T,\eta}^{\mathrm{P}} - \frac{1}{\tilde{\rho}}\mathcal{E}_T^{\mathrm{D}} \\
&\ge \sum_{t=1}^T f_t(\boldsymbol{\xi}^*) - T^{1/4}\mathcal{E}_{T,\eta} - \left(1 + 2T^{1/4}\right)\mathcal{E}_{T,\eta}^{\mathrm{P}} - T^{1/4}\mathcal{E}_T^{\mathrm{D}}.
\end{aligned}
$$

These steps are similar to those used to prove the regret bound in Theorem 4.6 (see the proof of Theorem 4.6 for further details). By recalling that $\boldsymbol{\xi}^*$ is an optimal solution to Problem $\mathrm{LP}_{\bar{f},\bar{g}}$ and $R^T = T\,\mathrm{OPT}_{\bar{f},\bar{g}} - \sum_{t=1}^{T} f_t(\boldsymbol{x}_t)$, the following bound on the cumulative regret holds:

$$
R^T = \sum_{t=1}^{T} f_t(\boldsymbol{\xi}^*) - \sum_{t=1}^{T} f_t(\boldsymbol{x}_t) \le T^{1/4}\mathcal{E}_{T,\eta} + \left(1 + 2T^{1/4}\right)\mathcal{E}_{T,\eta}^{\mathrm{P}} + T^{1/4}\mathcal{E}_{T}^{\mathrm{D}}.
$$

Next, we provide an upper bound on the cumulative constraints violation.

For every $i \in [m]$, the following holds

$$
\sum_{t=T_1+1}^{T} g_{t,i}(\boldsymbol{x}_t) \le -(T - T_1)\rho + 2\mathcal{E}_{T-T_1,\eta}^{\mathrm{P}} + \mathcal{E}_{T-T_1}^{\mathrm{D}} + \mathcal{E}_{T-T_1,\eta}
$$

$$
\le 2\mathcal{E}_{T-T_1,\eta}^{\mathrm{P}} + \mathcal{E}_{T-T_1}^{\mathrm{D}} + \mathcal{E}_{T-T_1,\eta}, \tag{15}
$$

where the first inequality follows from Lemma 4.4, while the second one from $\rho \ge 0$. Hence, for every $i \in [m]$, it holds

$$
\begin{aligned}
\sum_{t=1}^{T} g_{t,i}(\boldsymbol{x}_t) &= \sum_{t=1}^{T_1} g_{t,i}(\boldsymbol{x}_t) + \sum_{t=T_1+1}^{T} g_{t,i}(\boldsymbol{x}_t) \\
&\le (T - T_1)\tilde{\rho} + M_{\tilde{\rho}} + 2\mathcal{E}_{T-T_1,\eta}^{\mathrm{P}} + \mathcal{E}_{T-T_1}^{\mathrm{D}} + \mathcal{E}_{T-T_1,\eta} \\
&\le (T - T_1)T^{-1/4} + M_{T^{-1/4}} + 2\mathcal{E}_{T-T_1,\eta}^{\mathrm{P}} + \mathcal{E}_{T-T_1}^{\mathrm{D}} + \mathcal{E}_{T-T_1,\eta} \\
&\le T^{3/4} + M_{T^{-1/4}} + 2\mathcal{E}_{T-T_1,\eta}^{\mathrm{P}} + \mathcal{E}_{T-T_1}^{\mathrm{D}} + \mathcal{E}_{T-T_1,\eta} \\
&\le T^{3/4} + M_{T^{-1/4}} + 2\mathcal{E}_{T,\eta}^{\mathrm{P}} + \mathcal{E}_{T}^{\mathrm{D}} + \mathcal{E}_{T,\eta}.
\end{aligned}
$$

The first inequality follows from Equation (15) and from the condition in Line 4 of Algorithm 1, which ensures that $\sum_{t=1}^{T_1} g_{t,i}(\boldsymbol{x}_t) \le (T - T_1)\tilde{\rho} + M_{\tilde{\rho}}$ for every $i \in [m]$. Moreover, the second inequality follows from $\tilde{\rho} = T^{-1/4}$. Thus, by letting $i^\star \in \arg\max_{i \in [m]} \sum_{t=1}^{T} g_{t,i}(\boldsymbol{x}_t)$, and by recalling that $V^T = \max_{i \in [m]} \sum_{t=1}^{T} g_{t,i}(\boldsymbol{x}_t)$, the following bound on the cumulative constraints violation holds:

$$
V^T = \sum_{t=1}^{T} g_{t,i^\star}(\boldsymbol{x}_t) \le T^{3/4} + M_{T^{-1/4}} + 2\mathcal{E}_{T,\eta}^{\mathrm{P}} + \mathcal{E}_{T}^{\mathrm{D}} + \mathcal{E}_{T,\eta}.
$$

This concludes the proof. □

## B.3 Proof omitted from Section 5

First, we provide a preliminary result on the value of the Lagrangian game when primal and dual players are constrained to specific sets of strategies.

**Lemma B.1.** *Let* $f \in \mathcal{F}$ *and* $g \in \mathcal{G}$ *be such that* $d_g > 0$. *Moreover, given any* $\epsilon > 0$, *let* $\Xi_{\epsilon,g} := \left\{ \boldsymbol{\xi} \in \Xi : \max_{i \in [m]} g_i(\boldsymbol{\xi}) \ge \epsilon \right\}$. *The following holds:*

$$
\sup_{\boldsymbol{\xi} \in \Xi_\epsilon} \inf_{\boldsymbol{\lambda} \in \mathcal{D}_{d_g/2}} \mathcal{L}_{f,g}(\boldsymbol{\xi}, \boldsymbol{\lambda}) \le \mathrm{OPT}_{f,g} - \frac{\epsilon}{d_g}.
$$

*Proof.* Let $\boldsymbol{\xi} \in \Xi_{\epsilon,g}$ and $i^\star \in \arg\max_{i \in [m]} g_i(\boldsymbol{\xi})$. Then,

$$
\begin{aligned}
\inf_{\boldsymbol{\lambda} \in \mathcal{D}_{d_g/2}} \left\{ f(\boldsymbol{\xi}) - \langle \boldsymbol{\lambda}, g(\boldsymbol{\xi}) \rangle \right\} &= f(\boldsymbol{\xi}) - \frac{2}{d_g} g_{i^\star}(\boldsymbol{\xi}) \\
&= \inf_{\boldsymbol{\lambda} \in \mathcal{D}_{d_g}} \left\{ f(\boldsymbol{\xi}) - \langle \boldsymbol{\lambda}, g(\boldsymbol{\xi}) \rangle \right\} - \frac{1}{d_g} g_{i^\star}(\boldsymbol{\xi}) \\
&\leq \sup_{\boldsymbol{\xi} \in \Xi} \inf_{\boldsymbol{\lambda} \in \mathcal{D}_{d_g}} \mathcal{L}_{f,g}(\boldsymbol{\xi}, \boldsymbol{\lambda}) - \frac{1}{d_g} g_{i^\star}(\boldsymbol{\xi}) \\
&\leq \mathrm{OPT}_{f,g} - \frac{1}{d_g} g_{i^\star}(\boldsymbol{\xi}) \\
&\leq \mathrm{OPT}_{f,g} - \frac{\epsilon}{d_g},
\end{aligned}
$$

where the second inequality follows from Theorem 2.3, while the last one holds by the definition of $\Xi_{\epsilon,g}$ and $i^\star$. $\square$

Next, we introduce a new event that extends $\mathbf{E}$ by considering also the (stochastic) sequence of reward functions $f_t$. Formally, the event is defined as follows.

**Definition B.2.** We denote with $\bar{\mathbf{E}}$ the event in which Algorithm 1 satisfies the following conditions (recall that $\eta = \delta/3$): (i) event $\mathbf{E}$ holds; (ii) for every pair of rounds $t, t' \in [T] : t \leq t'$ it holds:

- $|\sum_{\tau=t}^{t'} f_\tau(\boldsymbol{x}_\tau) - \sum_{\tau=t}^{t'} \bar{f}(\boldsymbol{x}_\tau)| \leq \mathcal{E}_{t'-t,\eta}$,

- $|\sum_{\tau=t}^{t'} f_\tau(\boldsymbol{\xi}^*) - \sum_{\tau=t}^{t'} \bar{f}(\boldsymbol{\xi}^*)| \leq \mathcal{E}_{t'-t,\eta}$.

**Lemma B.3.** *After running Algorithm 1, the event $\bar{\mathbf{E}}$ holds with probability at least $1 - \delta$.*

*Proof.* Given a desired failure probability $\delta \in (0,1)$, recall that $\eta = \delta/3$ and set $\varepsilon = \eta/12mT^2$. Consider the following inequalities in which the differences between expectations and empirical means of reward functions are bounded:

$$
\left| \sum_{\tau=t}^{t'} f_\tau(\boldsymbol{x}_\tau) - \sum_{\tau=t}^{t'} \bar{f}(\boldsymbol{x}_\tau) \right| > 2\sqrt{2(t'-t)\ln\frac{1}{\varepsilon}}, \tag{16}
$$

$$
\left| \sum_{\tau=t}^{t'} f_\tau(\boldsymbol{\xi}^*) - \sum_{\tau=t}^{t'} \bar{f}(\boldsymbol{\xi}^*) \right| > 2\sqrt{2(t'-t)\ln\frac{1}{\varepsilon}}. \tag{17}
$$

By applying the Azuma-Hoeffding inequality to each martingale difference sequence, we get that each inequality holds with probability at most $2\varepsilon$. We denote by $\bar{\mathbf{E}}_\eta$ the event in which Equations (16) and (17) hold for every $t \leq t' \in [T] : t < t'$ and event $\mathbf{E}_\eta$ holds (see the proof of Lemma 4.2 for the definition of event $\mathbf{E}_\eta$). By a union bound, we have that:

$$
\mathbb{P}(\bar{\mathbf{E}}_\eta) \geq 1 - 2\varepsilon(4mT^2 + 2T^2) \geq 1 - \eta.
$$

Therefore, event $\bar{\mathbf{E}}_\eta$ holds with probability at least $1 - \eta$. Moreover, let us recall that:

$$
\mathcal{E}_{t'-t,\eta} = \sqrt{8(t'-t)\ln\frac{12mT^2}{\eta}} = 2\sqrt{2(t'-t)\ln\frac{1}{\varepsilon}}.
$$

Now, consider the event $\bar{\mathbf{E}}$ in which Algorithm 1 satisfies the following conditions: (i) the regret incurred by $\mathcal{R}_{\mathrm{I}}^{\mathrm{P}}$ after $T_1$ rounds is upper bounded by $\mathcal{E}_{T_1,\eta}^{\mathrm{P}}$; (ii) the regret cumulated by $\mathcal{R}_{\mathrm{II}}^{\mathrm{P}}$ after the remaining $T - T_1$ rounds is upper bounded by $\mathcal{E}_{T-T_1,\eta}^{\mathrm{P}}$; and (iii) event $\bar{\mathbf{E}}_\eta$ holds. Recall that each one of the conditions (i), (ii) and (iii) holds with probability at least $1 - \eta$; hence, by a union bound we get:

$$
\mathbb{P}(\bar{\mathbf{E}}) \geq 1 - 3\eta = 1 - \delta.
$$

This concludes the proof. $\square$

As a first step, we prove that the primal regret minimizer gets a per-round utility that is close to the value $\mathrm{OPT}_{\bar{f},\bar{g}}$. Formally:

**Lemma B.4.** *If the event $\bar{\mathbf{E}}$ holds, then, for every round $\tau \in [T_1]$ the following inequality holds:*

$$\sum_{t=1}^{\tau} \mathcal{L}_{f_t,g_t}(\boldsymbol{x}_t, \boldsymbol{\lambda}_t) \geq \tau \, \mathrm{OPT}_{\bar{f},\bar{g}}^{\mathrm{LP}} - \left(1 + \frac{2}{\tilde{\rho}}\right)\mathcal{E}_{\tau,\eta}^{\mathrm{P}} - \left(1 + \frac{1}{\tilde{\rho}}\right)\mathcal{E}_{\tau,\eta}.$$

*Proof.* Let $\boldsymbol{\xi}^{\star}$ be an optimal solution to Problem $\mathrm{LP}_{\bar{f},\bar{g}}$, and let $\bar{\boldsymbol{\lambda}} = \frac{1}{\tau}\sum_{t=1}^{\tau} \boldsymbol{\lambda}_t$. Then, it holds

$$\sum_{t=1}^{\tau} \mathcal{L}_{f_t,g_t}(\boldsymbol{x}_t, \boldsymbol{\lambda}_t) \geq \sum_{t=1}^{\tau} \mathcal{L}_{f_t,g_t}(\boldsymbol{\xi}^*, \boldsymbol{\lambda}_t) - \left(1 + \frac{2}{\tilde{\rho}}\right)\mathcal{E}_{\tau,\eta}^{\mathrm{P}}$$

$$\geq \sum_{t=1}^{\tau} \mathcal{L}_{\bar{f},\bar{g}}(\boldsymbol{\xi}^*, \boldsymbol{\lambda}_t) - \left(1 + \frac{2}{\tilde{\rho}}\right)\mathcal{E}_{\tau,\eta}^{\mathrm{P}} - \left(1 + \frac{1}{\tilde{\rho}}\right)\mathcal{E}_{\tau,\eta}$$

$$= \sum_{t=1}^{\tau} \mathcal{L}_{\bar{f},\bar{g}}(\boldsymbol{\xi}^*, \bar{\boldsymbol{\lambda}}) - \left(1 + \frac{2}{\tilde{\rho}}\right)\mathcal{E}_{\tau,\eta}^{\mathrm{P}} - \left(1 + \frac{1}{\tilde{\rho}}\right)\mathcal{E}_{\tau,\eta}$$

$$\geq \tau \inf_{\boldsymbol{\lambda} \in \mathcal{D}_{\tilde{\rho}}} \mathcal{L}_{\bar{f},\bar{g}}(\boldsymbol{\xi}^*, \boldsymbol{\lambda}) - \left(1 + \frac{2}{\tilde{\rho}}\right)\mathcal{E}_{\tau,\eta}^{\mathrm{P}} - \left(1 + \frac{1}{\tilde{\rho}}\right)\mathcal{E}_{\tau,\eta}$$

$$= \tau \sup_{\boldsymbol{\xi} \in \Xi} \inf_{\boldsymbol{\lambda} \in \mathcal{D}_{\tilde{\rho}}} \mathcal{L}_{\bar{f},\bar{g}}(\boldsymbol{\xi}, \boldsymbol{\lambda}) - \left(1 + \frac{2}{\tilde{\rho}}\right)\mathcal{E}_{\tau,\eta}^{\mathrm{P}} - \left(1 + \frac{1}{\tilde{\rho}}\right)\mathcal{E}_{\tau,\eta}$$

$$= \tau \mathrm{OPT}_{\bar{f},\bar{g}}^{\mathrm{LP}} - \left(1 + \frac{2}{\tilde{\rho}}\right)\mathcal{E}_{\tau,\eta}^{\mathrm{P}} - \left(1 + \frac{1}{\tilde{\rho}}\right)\mathcal{E}_{\tau,\eta},$$

where the first inequality follows from the no-regret property of the primal regret minimizer, the second one from the definition of the event $\bar{\mathbf{E}}$, and the third one from the definition of $\boldsymbol{\xi}^*$. Moreover, the first equality follows from the fact that $\bar{g}$ is independent from $t$. This concludes the proof. $\square$

Now, we show that the dual regret minimizer gets a per-round utility that is close to the value $\mathrm{OPT}_{\bar{f},\bar{g}}$. Moreover, the attained utility increases by an additive factor proportional to the primal violation. This can be proved only in the setting with stochastic reward functions. Indeed, in this setting the primal and dual regret minimizers are playing a stochastic repeated zero-sum game that converges to an equilibrium. Notice that this is *not* true when the reward functions are adversarial.

**Lemma B.5.** *If event $\bar{\mathbf{E}}$ holds and Condition 4.5 is satisfied, then for each $\tau \in [T_1]$ and each $i \in [m]$*

$$\sum_{t=1}^{\tau} \mathcal{L}_{f_t,g_t}(\boldsymbol{\xi}_t, \boldsymbol{\lambda}_t) \leq \tau \mathrm{OPT}_{\bar{f},\bar{g}}^{\mathrm{LP}} + \frac{1}{\tilde{\rho}}\mathcal{E}_{\tau}^{\mathrm{D}} + \left(1 + \frac{2}{\tilde{\rho}}\right)\mathcal{E}_{\tau,\eta} - \sum_{t=1}^{\tau} g_{t,i}(\boldsymbol{x}_t).$$

*Proof.* In the following, let $\boldsymbol{\lambda}^* \in \arg\min_{\boldsymbol{\lambda} \in \mathcal{D}_{\tilde{\rho}}} \sum_{t=1}^{\tau} \mathcal{L}_{\bar{f},\bar{g}}(\boldsymbol{\xi}_t, \boldsymbol{\lambda})$, $\epsilon := \frac{\max_{i \in [m]} \sum_{t=1}^{\tau} g_{t,i}(\boldsymbol{x}_t) - \mathcal{E}_{\tau,\eta}}{\tau}$, and $\bar{\boldsymbol{\xi}} := \frac{1}{\tau}\sum_{t=1}^{\tau} \boldsymbol{\xi}_t$, where $\boldsymbol{\xi}_t \in \Xi$ denotes the strategy mixture that plays deterministically $\boldsymbol{x}_t$. Moreover, let us define the set $\Xi_{\epsilon,\bar{g}} := \{\boldsymbol{\xi} \in \Xi : \max_{i \in [m]} \bar{g}_i(\boldsymbol{\xi}) \geq \epsilon\}$.

As a first step, we prove that $\bar{\boldsymbol{\xi}} \in \Xi_{\epsilon,\bar{g}}$. In particular, since the event $\bar{\mathbf{E}}$ holds, we have that

$$\max_{i \in [m]} \bar{g}_i(\bar{\boldsymbol{\xi}}) \geq \frac{\sum_{t=1}^{\tau} \max_{i \in [m]} g_i(\bar{\boldsymbol{\xi}}) - \mathcal{E}_{\tau,\eta}}{\tau} = \epsilon$$

For every $\tau \in [T_1]$, we have:

$$\sum_{t=1}^{\tau} \mathcal{L}_{f_t,g_t}(\boldsymbol{x}_t, \boldsymbol{\lambda}_t) \leq \sum_{t=1}^{\tau} \mathcal{L}_{f_t,g_t}(\boldsymbol{x}_t, \boldsymbol{\lambda}^*) + \frac{1}{\tilde{\rho}}\mathcal{E}_{\tau}^{\mathrm{D}} \tag{18a}$$

$$\leq \sum_{t=1}^{\tau} \mathcal{L}_{\bar{f},\bar{g}}(\boldsymbol{x}_t, \boldsymbol{\lambda}^*) + \frac{1}{\tilde{\rho}}\mathcal{E}_{\tau}^{\mathrm{D}} + \left(1 + \frac{1}{\tilde{\rho}}\right)\mathcal{E}_{\tau,\eta} \tag{18b}$$

$$\leq \inf_{\boldsymbol{\lambda}\in\mathcal{D}_{\tilde{\rho}}} \sum_{t=1}^{\tau} \mathcal{L}_{\bar{f},\bar{g}}(\boldsymbol{x}_t, \boldsymbol{\lambda}) + \frac{1}{\tilde{\rho}}\mathcal{E}_{\tau}^{\mathrm{D}} + \left(1+\frac{1}{\tilde{\rho}}\right)\mathcal{E}_{\tau,\eta} \tag{18c}$$

$$= \tau \inf_{\boldsymbol{\lambda}\in\mathcal{D}_{\tilde{\rho}}} \mathcal{L}_{\bar{f},\bar{g}}(\bar{\boldsymbol{\xi}}, \boldsymbol{\lambda}) + \frac{1}{\tilde{\rho}}\mathcal{E}_{\tau}^{\mathrm{D}} + \left(1+\frac{1}{\tilde{\rho}}\right)\mathcal{E}_{\tau,\eta} \tag{18d}$$

$$= \tau \inf_{\boldsymbol{\lambda}\in\mathcal{D}_{\rho/2}} \mathcal{L}_{\bar{f},\bar{g}}(\bar{\boldsymbol{\xi}}, \boldsymbol{\lambda}) + \frac{1}{\tilde{\rho}}\mathcal{E}_{\tau}^{\mathrm{D}} + \left(1+\frac{1}{\tilde{\rho}}\right)\mathcal{E}_{\tau,\eta} \tag{18e}$$

$$\leq \tau \sup_{\boldsymbol{\xi}\in\Xi_{\epsilon,\bar{g}}} \inf_{\boldsymbol{\lambda}\in\mathcal{D}_{\rho/2}} \mathcal{L}_{\bar{f},\bar{g}}(\boldsymbol{\xi}, \boldsymbol{\lambda}) + \frac{1}{\tilde{\rho}}\mathcal{E}_{\tau}^{\mathrm{D}} + \left(1+\frac{1}{\tilde{\rho}}\right)\mathcal{E}_{\tau,\eta} \tag{18f}$$

$$\leq \tau \sup_{\boldsymbol{\xi}\in\Xi} \inf_{\boldsymbol{\lambda}\in\mathcal{D}_{\rho}} \left(\mathcal{L}_{\bar{f},\bar{g}}(\boldsymbol{\xi}, \boldsymbol{\lambda}) - \frac{\epsilon}{\rho}\right) + \frac{1}{\tilde{\rho}}\mathcal{E}_{\tau}^{\mathrm{D}} + \left(1+\frac{1}{\tilde{\rho}}\right)\mathcal{E}_{\tau,\eta} \tag{18g}$$

$$= \tau\left(\mathrm{OPT}_{\bar{f},\bar{g}} - \frac{\epsilon}{\rho}\right) + \frac{1}{\tilde{\rho}}\mathcal{E}_{\tau}^{\mathrm{D}} + \left(1+\frac{1}{\tilde{\rho}}\right)\mathcal{E}_{\tau,\eta} \tag{18h}$$

$$= \tau\mathrm{OPT}_{\bar{f},\bar{g}} - \tau\frac{\max_{i'\in[m]}\sum_{t=1}^{\tau} g_{t,i'}(\boldsymbol{x}_t) - \mathcal{E}_{\tau,\eta}}{\tau\rho} + \frac{1}{\tilde{\rho}}\mathcal{E}_{\tau}^{\mathrm{D}} + \left(1+\frac{1}{\tilde{\rho}}\right)\mathcal{E}_{\tau,\eta} \tag{18i}$$

$$\leq \tau\mathrm{OPT}_{\bar{f},\bar{g}} + \frac{1}{\tilde{\rho}}\mathcal{E}_{\tau}^{\mathrm{D}} + \left(1+\frac{2}{\tilde{\rho}}\right)\mathcal{E}_{\tau,\eta} - \max_{i'\in[m]}\frac{\sum_{t=1}^{\tau} g_{t,i'}(2\boldsymbol{x}_t)}{\rho} \tag{18j}$$

$$\leq \tau\mathrm{OPT}_{\bar{f},\bar{g}} + \frac{1}{\tilde{\rho}}\mathcal{E}_{\tau}^{\mathrm{D}} + \left(1+\frac{2}{\tilde{\rho}}\right)\mathcal{E}_{\tau,\eta} - \max_{i'\in[m]}\sum_{t=1}^{\tau} g_{t,i'}(\boldsymbol{x}_t) \tag{18k}$$

$$\leq \tau\mathrm{OPT}_{\bar{f},\bar{g}} + \frac{1}{\tilde{\rho}}\mathcal{E}_{\tau}^{\mathrm{D}} + \left(1+\frac{2}{\tilde{\rho}}\right)\mathcal{E}_{\tau,\eta} - \sum_{t=1}^{\tau} g_{t,i}(\boldsymbol{x}_t) \qquad \forall i \in [m], \tag{18l}$$

where Equation (18a) is given by the no-regret property of the dual regret minimizer, and Equation (18b) by the definition of the event $\bar{\mathbf{E}}$, which holds by assumption. Moreover, Equation (18d) follows from the fact that $\bar{f}$ and $\bar{g}$ are independent from $t$, Equation (18e) follows from $\tilde{\rho} = \hat{\rho}/2 \leq \rho/2$, and Equation (18f) from $\bar{\boldsymbol{\xi}} \in \Xi_{\epsilon,\bar{g}}$. Finally, Equation (18g) follows from Lemma B.1, Equation (18i) by definition of $\epsilon$, and Equation (18j) by $\tilde{\rho} \leq \rho$. $\qquad\square$

**Theorem 5.1.** *Suppose that functions $f_t$ and $g_t$ are selected stochastically. With probability at least $1 - \delta$, Algorithm 1 never enters the recovery phase, namely $T_1 = T$.*

*Proof.* We prove the statement of the theorem by considering two cases.

**Case "Condition 4.5 holds".** By Lemma 4.2, event $\mathbf{E}$ holds with probability at least $1 - \delta$. In the rest of the proof, we assume that the event $\mathbf{E}$ holds, and we provide a bound that holds with probability at least $1 - \delta$. For every $\tau \in [T_1]$, we have:

$$\begin{aligned}
\sum_{t=1}^{\tau} g_t(\boldsymbol{x}_t) &\leq \tau\mathrm{OPT}_{\bar{f},\bar{g}} - \sum_{t=1}^{\tau} \mathcal{L}_{f_t,g_t}(\boldsymbol{x}_t, \boldsymbol{\lambda}_t) + \frac{1}{\tilde{\rho}}\mathcal{E}_{\tau}^{\mathrm{D}} + \left(1+\frac{2}{\tilde{\rho}}\right)\mathcal{E}_{\tau,\eta} \\
&\leq \left(1+\frac{2}{\tilde{\rho}}\right)\mathcal{E}_{\tau,\eta}^{\mathrm{P}} + \left(1+\frac{1}{\tilde{\rho}}\right)\mathcal{E}_{\tau,\eta} + \frac{1}{\tilde{\rho}}\mathcal{E}_{\tau}^{\mathrm{D}} + \left(1+\frac{2}{\tilde{\rho}}\right)\mathcal{E}_{\tau,\eta} \\
&= \left(2+\frac{3}{\tilde{\rho}}\right)\mathcal{E}_{\tau,\eta} + \left(1+\frac{2}{\tilde{\rho}}\right)\mathcal{E}_{\tau,\eta}^{\mathrm{P}} + \frac{1}{\tilde{\rho}}\mathcal{E}_{\tau}^{\mathrm{D}} \\
&\leq \frac{2}{\tilde{\rho}}\sqrt{T} - 1 + \left(2+\frac{3}{\tilde{\rho}}\right)\mathcal{E}_{\tau,\eta} + \left(1+\frac{2}{\tilde{\rho}}\right)\mathcal{E}_{\tau,\eta}^{\mathrm{P}} + \frac{1}{\tilde{\rho}}\mathcal{E}_{\tau}^{\mathrm{D}} \\
&= M_{\tilde{\rho}} - 1,
\end{aligned}$$

where the first inequality follows from Lemma B.4, the second one from Lemma B.5, the third one from the fact that $\frac{2}{\tilde{\rho}}\sqrt{T} - 1 \geq 0$, being $\tilde{\rho} \leq 1$, and the last equation follows from the definition of $M_{\tilde{\rho}}$. This implies that the algorithm never enters the recovery phase when Condition 4.5 holds.

**Case "Condition 4.5 does *not* hold".** By Lemma 4.2, event **E** holds with probability at least $1 - \delta$. In the rest of the proof, we assume that the event **E** holds, and we provide a bound that holds with probability at least $1 - \delta$. Suppose by contradiction that $T_1 < T$. This implies that a constraint $i \in [m]$ is violated by at least $M_{T^{-1/4}} - 1$. Let $i^\star \in \arg\max_{i \in [m]} \sum_{t=1}^{T_1} g_{t,i}(\boldsymbol{x}_t)$ be one of the most violated constraints during the play phase. Then, we have:

$$
\sum_{t=1}^{T_1} \mathcal{L}_{f_t, g_t}(\boldsymbol{x}_t, \boldsymbol{\lambda}_t) = \sum_{t=1}^{T_1} \Big( f(\boldsymbol{x}_t) - \langle \boldsymbol{\lambda}_t, g_t(\boldsymbol{x}_t) \rangle \Big)
$$

$$
\leq T_1 - \sum_{t=1}^{T_1} \langle \boldsymbol{\lambda}_t, g_t(\boldsymbol{x}_t) \rangle
$$

$$
\leq T_1 - \sum_{t=1}^{T_1} \frac{1}{T^{-1/4}} g_{t,i^\star}(\boldsymbol{x}_t) + T^{1/4} \mathcal{E}^{\mathrm{D}}_{T_1}
$$

$$
\leq T_1 - T^{1/4}(M_{T^{-1/4}} - 1) + T^{1/4} \mathcal{E}^{\mathrm{D}}_{\tau_1}
$$

$$
< -\Big(1 + \frac{2}{T^{-1/4}}\Big) \mathcal{E}^{\mathrm{P}}_{\tau, \eta} - \frac{1}{T^{-1/4}} \mathcal{E}_{\tau, \eta},
$$

where the second inequality follows from the no-regret property of the dual regret minimizer and the fact that, when Condition 4.5 does *not* hold, $\tilde{\rho} = T^{-1/4}$. The last inequality follows from the definition of $M_{T^{-1/4}}$. Then, the result above allows us to reach the desired contradiction when compared with the following one. In particular, for every $\tau \in [T_1]$, we have:

$$
\sum_{t=1}^{\tau} \mathcal{L}_{f_t, g_t}(\boldsymbol{x}_t, \boldsymbol{\lambda}_t) \geq \sum_{t=1}^{\tau} \mathcal{L}_{f_t, g_t}(\boldsymbol{\xi}^\circ, \boldsymbol{\lambda}_t) - \Big(1 + \frac{2}{T^{-1/4}}\Big) \mathcal{E}^{\mathrm{P}}_{\tau, \eta}
$$

$$
\geq \sum_{t=1}^{\tau} \mathcal{L}_{f_t, \bar{g}}(\boldsymbol{\xi}^\circ, \boldsymbol{\lambda}_t) - \frac{1}{T^{-1/4}} \mathcal{E}_{\tau, \eta} - \Big(1 + \frac{2}{T^{-1/4}}\Big) \mathcal{E}^{\mathrm{P}}_{\tau, \eta}
$$

$$
\geq -\frac{1}{T^{-1/4}} \mathcal{E}_{\tau, \eta} - \Big(1 + \frac{2}{T^{-1/4}}\Big) \mathcal{E}^{\mathrm{P}}_{\tau, \eta},
$$

where the first inequality follows from the no-regret property of the primal regret minimizer, the second one follows from the fact that event **E** holds, and the third one from the feasibility of $\boldsymbol{\xi}^\circ$. □

## B.4 Proofs omitted from Section 6

As a first step, we provide a lower bound for the cumulative reward achieved during the play phase. In particular, we show that it achieves at least a $\rho/(1 + \rho)$ fraction of the value obtained by an optimal solution in the first $T_1$ rounds.

**Lemma B.6.** *If Condition* 4.5 *is satisfied, then, with probability at least* $1 - \eta$, *at round* $T_1$ *of Algorithm 1 it holds that:*

$$
\sum_{t=1}^{T_1} f_t(\boldsymbol{x}_t) \geq \frac{\rho}{1 + \rho} \sum_{t=1}^{T_1} f_t(\boldsymbol{\xi}^*) + (T - T_1) - \Big(1 + \frac{2}{\tilde{\rho}}\Big) \mathcal{E}^{\mathrm{P}}_{T_1, \eta} - \frac{1}{\tilde{\rho}} \mathcal{E}^{\mathrm{D}}_{\tau_1}.
$$

*Proof.* Let $\bar{\boldsymbol{\xi}} \in \Xi$ be a strategy mixture obtained by playing with probability $1/(1 + \rho)$ the mixture $\boldsymbol{\xi}^\circ$ and with the remaining probability $\rho/(1 + \rho)$ an optimal mixture $\boldsymbol{\xi}^*$. Notice that the probabilities are well defined, since $\rho \geq 0$. Then, for every $t \in [T]$ and $i \in [m]$, it holds:

$$
\frac{1}{1 + \rho} g_{t,i}(\boldsymbol{\xi}^\circ) + \frac{\rho}{1 + \rho} g_{t,i}(\boldsymbol{\xi}^*) \leq -\frac{\rho}{1 + \rho} + \frac{\rho}{1 + \rho} = 0
$$

where the inequality follows from the fact that $g_{t,i}(\boldsymbol{\xi}^\circ) \leq -\rho$ and $g_{t,i}(\boldsymbol{\xi}^*) \leq 1$. Therefore, for every $t \in [T]$ and $i \in [m]$, it holds that $g_t(\bar{\boldsymbol{\xi}}) \leq 0$. Assume that the regret bounds of the regret minimizers hold. Notice that this happens with probability at least $1 - \eta$. Then, by the no-regret property of the

primal regret minimizer, we have that

$$\sum_{t=1}^{T_1} \mathcal{L}_{f_t,g_t}(\boldsymbol{x}_t, \boldsymbol{\lambda}_t) \geq \sum_{t=1}^{T_1} \mathcal{L}_{f_t,g_t}(\bar{\boldsymbol{\xi}}, \boldsymbol{\lambda}_t) - \left(1 + \frac{2}{\tilde{\rho}}\right)\mathcal{E}_{T_1,\eta}^{\mathbb{P}}. \tag{19}$$

Let $i^\star \in \arg\max_{i\in[m]} \sum_{t=1}^{T_1} g_{t,i}(\boldsymbol{x}_t)$ be one of the most violated constraints during the play phase. Next, we prove that

$$\sum_{t=1}^{T_1} \langle \boldsymbol{\lambda}_t, g_t(\boldsymbol{x}_t)\rangle \geq (T - T_1) - \frac{1}{\tilde{\rho}}\mathcal{E}_{T_1}^{\mathbb{D}}.$$

We consider two cases. If $T_1 = T$, then

$$\sum_{t=1}^{T_1} \langle \boldsymbol{\lambda}_t, g_t(\boldsymbol{x}_t)\rangle \geq \sum_{t=1}^{T_1} \langle \boldsymbol{0}, g_t(\bar{\boldsymbol{\xi}})\rangle - \frac{1}{\tilde{\rho}}\mathcal{E}_{T_1}^{\mathbb{D}} = -\frac{1}{\tilde{\rho}}\mathcal{E}_{\tau_2}^{\mathbb{D}} = (T - T_1) - \frac{1}{\tilde{\rho}}\mathcal{E}_{T_1}^{\mathbb{D}}.$$

Otherwise, we have that $\sum_{t=1}^{T_1} g_{t,i^\star}(\boldsymbol{x}_t) \geq \tilde{\rho}(T - T_1)$ and

$$\sum_{t=1}^{T_1} \langle \boldsymbol{\lambda}_t, g_t(\boldsymbol{x}_t)\rangle \geq \left(\sum_{t=1}^{T_1} \frac{1}{\tilde{\rho}} g_{t,i^\star}(\boldsymbol{x}_t)\right) - \frac{1}{\tilde{\rho}}\mathcal{E}_{T_1}^{\mathbb{D}} \geq (T - T_1) - \frac{1}{\tilde{\rho}}\mathcal{E}_{T_1}^{\mathbb{D}}. \tag{20}$$

Thus,

$$\begin{aligned}
\sum_{t=1}^{T_1} f_t(\boldsymbol{x}_t) &\geq \sum_{t=1}^{T_1} \left(f_t(\bar{\boldsymbol{\xi}}) - \langle \boldsymbol{\lambda}_t, g_t(\bar{\boldsymbol{\xi}})\rangle + \langle \boldsymbol{\lambda}_t, g_t(\boldsymbol{x}_t)\rangle\right) - \left(1 + \frac{2}{\tilde{\rho}}\right)\mathcal{E}_{T_1,\eta}^{\mathbb{P}} \\
&\geq \sum_{t=1}^{T_1} \left(f_t(\bar{\boldsymbol{\xi}}) - \langle \boldsymbol{\lambda}_t, g_t(\bar{\boldsymbol{\xi}})\rangle\right) + (T - T_1) - \left(1 + \frac{2}{\tilde{\rho}}\right)\mathcal{E}_{T_1,\eta}^{\mathbb{P}} - \frac{1}{\tilde{\rho}}\mathcal{E}_{\tau_2}^{\mathbb{D}} \\
&\geq \sum_{t=1}^{T_1} f_t(\bar{\boldsymbol{\xi}}) + (T - T_1) - \left(1 + \frac{2}{\tilde{\rho}}\right)\mathcal{E}_{T_1,\eta}^{\mathbb{P}} - \frac{1}{\tilde{\rho}}\mathcal{E}_{\tau_2}^{\mathbb{D}} \\
&\geq \sum_{t=1}^{T_1} \left(\frac{1}{1+\rho} f_t(\boldsymbol{\xi}^\circ) + \frac{\rho}{1+\rho} f_t(\boldsymbol{\xi}^*)\right) + (T - T_1) - \left(1 + \frac{2}{\tilde{\rho}}\right)\mathcal{E}_{T_1,\eta}^{\mathbb{P}} - \frac{1}{\tilde{\rho}}\mathcal{E}_{\tau_2}^{\mathbb{D}} \\
&\geq \frac{\rho}{1+\rho} \sum_{t=1}^{T_1} f_t(\boldsymbol{\xi}^*) + (T - T_1) - \left(1 + \frac{2}{\tilde{\rho}}\right)\mathcal{E}_{T_1,\eta}^{\mathbb{P}} - \frac{1}{\tilde{\rho}}\mathcal{E}_{T_1}^{\mathbb{D}},
\end{aligned}$$

where the first inequality follows from Equation (19), the second one from Equation 20, the third one from the fact that for each $t \in [T]$ it holds $g_t(\bar{\boldsymbol{\xi}}) \leq 0$, while the fourth inequality follows from the definition of $\bar{\boldsymbol{\xi}}$. This concludes the proof. $\qquad\square$

Notice that, for a small $T_1$, we have a large lower bound on the cumulative reward. Intuitively, this means that when the play phase is short, the primal regret minimizer accumulated so much regret in the play phase that the recovery phase can be addressed without worrying about the reward.

As a second step, we provide an upper bound on the cumulative constraints violation during the recovery phase. In particular, we show that the constraints are satisfied by at least $\rho$ at each round up to a term related to the regret of $\mathcal{R}^{\mathbb{P}}$ and $\mathcal{R}^{\mathbb{D}}$.

**Lemma B.7.** *With probability at least $1 - \eta$, when Algorithm 1 halts it holds that for each $i \in [m]$:*

$$\sum_{t=T_1+1}^{T} g_{t,i}(\boldsymbol{x}_t) \leq -(T - T_1)\rho + \mathcal{E}_{T-T_1}^{\mathbb{D}} + 2\mathcal{E}_{T-T_1,\eta}^{\mathbb{P}}.$$

*Proof.* Let $i^\star$ be one of the most violated constraints, *i.e.*, $i^\star \in \arg\max_{i \in [m]} \sum_{t=T_1+1}^{T} g_{t,i}(\boldsymbol{x}_t)$. Then, we have that:

$$
\begin{aligned}
(T - \tau)\rho &\leq - \sum_{t=T_1+1}^{T} \langle \boldsymbol{\lambda}_t, g_t(\boldsymbol{\xi}^\circ) \rangle \\
&\leq - \sum_{t=T_1+1}^{T} \langle \boldsymbol{\lambda}_t, g_t(\boldsymbol{x}_t) \rangle + 2\mathcal{E}^{\mathrm{P}}_{T-T_1,\eta} \\
&\leq - \sum_{t=T_1+1}^{T} g_{t,i^\star}(\boldsymbol{x}_t) + \mathcal{E}^{\mathrm{D}}_{T-T_1} + 2\mathcal{E}^{\mathrm{P}}_{T-T_1,\eta},
\end{aligned}
$$

where the first inequality follows from the definition of $\boldsymbol{\xi}^\circ$ and the fact that it is always feasible of at least $\rho$, the second one follows from the assumption that the primal regret minimizer satisfies the regret bound, and the last inequality from the guarantee on the regret of the dual regret minimizer. We conclude the proof by noticing that the regret bound of the primal regret minimizer holds with probability at least $1 - \eta$. $\qquad\square$

Now, we can provide our bounds for adversarial constraints.

**Theorem 6.1.** *Suppose that functions $f_t$ and $g_t$ are selected adversarially. If Condition 4.5 is satisfied, then, with probability at least $1 - \frac{2}{3}\delta$, Algorithm 1 guarantees that the following holds:*
$$\sum_{t=1}^{T} f_t(\boldsymbol{x}_t) \geq \frac{\rho}{1+\rho} \sum_{t=1}^{T} \mathrm{OPT}_{\bar{f},\bar{g}} - \left(1 + \frac{2}{\tilde{\rho}}\right)\mathcal{E}^{\mathrm{P}}_{T,\eta} - \frac{1}{\tilde{\rho}}\mathcal{E}^{\mathrm{D}}_T \text{ and } V^T \leq M_{\tilde{\rho}} + 2\mathcal{E}^{\mathrm{P}}_{T,\eta} + \mathcal{E}^{\mathrm{D}}_T.$$

*Proof.* In the following, we assume that both Lemma B.6 and Lemma B.7. By an union bound, this holds with probability $1 - 2\eta = 1 - \frac{2}{3}\delta$. Then, it holds

$$
\begin{aligned}
\sum_{t=1}^{T} f_t(\boldsymbol{x}_t) &\geq \sum_{t=1}^{T_1} f_t(\boldsymbol{x}_t) \\
&\geq \sum_{t=1}^{T_1} \frac{\rho}{1+\rho} f_t(\boldsymbol{\xi}^\star) + (T - T_1) - \left(1 + \frac{2}{\tilde{\rho}}\right)\mathcal{E}^{\mathrm{P}}_{T_1,\eta} - \frac{1}{\tilde{\rho}}\mathcal{E}^{\mathrm{D}}_{T_1} \\
&\geq \frac{\rho}{1+\rho} \sum_{t=1}^{T} f_t(\boldsymbol{\xi}^\star) - \left(1 + \frac{2}{\tilde{\rho}}\right)\mathcal{E}^{\mathrm{P}}_{T_1,\eta} - \frac{1}{\tilde{\rho}}\mathcal{E}^{\mathrm{D}}_{T_1} \\
&\geq \frac{\rho}{1+\rho} \sum_{t=1}^{T} f_t(\boldsymbol{\xi}^\star) - \left(1 + \frac{2}{\tilde{\rho}}\right)\mathcal{E}^{\mathrm{P}}_{T,\eta} - \frac{1}{\tilde{\rho}}\mathcal{E}^{\mathrm{D}}_T,
\end{aligned}
$$

where the second inequality comes from Lemma B.6. This proves the bound on the regret.

By Lemma B.7, for each $i \in [m]$,

$$
\begin{aligned}
\sum_{t=1}^{T} g_{t,i}(\boldsymbol{x}_t) &= \sum_{t=1}^{T_1} g_{t,i}(\boldsymbol{x}_t) + \sum_{t=T_1+1}^{T} g_{t,i}(\boldsymbol{x}_t) \\
&\leq (T - T_1)\tilde{\rho} + M_{\tilde{\rho}} - (T - T_1)\rho + \mathcal{E}^{\mathrm{D}}_{T-T_1} + 2\mathcal{E}^{\mathrm{P}}_{T-T_1,\eta} \\
&\leq M_{\tilde{\rho}} + \mathcal{E}^{\mathrm{D}}_{T-T_1} + 2\mathcal{E}^{\mathrm{P}}_{T-T_1,\eta} \\
&\leq M_{\tilde{\rho}} + \mathcal{E}^{\mathrm{D}}_T + 2\mathcal{E}^{\mathrm{P}}_{T,\eta},
\end{aligned}
$$

where the second inequality comes from $\tilde{\rho} \leq \rho$.

$\qquad\square$

**Corollary 6.2.** *Suppose functions $f_t$ and $g_t$ are selected stochastically and adversarially, respectively. If Condition 4.5 is satisfied, then, with probability at least $1 - \delta$, Algorithm 1 provides $\sum_{t=1}^{T} f_t(\boldsymbol{x}_t) \geq \frac{\rho}{1+\rho} \sum_{t=1}^{T} \mathrm{OPT}_{\bar{f},\bar{g}} - \left(1 + \frac{2}{\tilde{\rho}}\right)\mathcal{E}^{\mathrm{P}}_{T,\eta} - \frac{1}{\tilde{\rho}}\mathcal{E}^{\mathrm{D}}_T - 2\mathcal{E}_{T,\eta}$ and $V^T \leq M_{\tilde{\rho}} + 2\mathcal{E}^{\mathrm{P}}_{T,\eta} + \mathcal{E}^{\mathrm{D}}_T + \mathcal{E}_{T,\eta}$.*

*Proof.* It is easy to see that Theorem 6.1 can be extended to consider the definition of $\boldsymbol{\xi}^\star$ for stochastic rewards. formally, it holds $\sum_{t=1}^T f_t(x_t) \geq \frac{\rho}{1+\rho} \sum_{t=1}^T f_t(\boldsymbol{\xi}^\star) - \left(1 + \frac{2}{\tilde{\rho}}\right) \mathcal{E}_{T,\eta}^{\text{P}} - \frac{1}{\tilde{\rho}} \mathcal{E}_T^{\text{D}}$.
Consider the two martingale difference sequences $\sum_{t=1}^T f_t(\boldsymbol{x}_t) - \bar{f}(\boldsymbol{x}_t)$ and $\sum_t f_t(\boldsymbol{\xi}^\star) - \bar{f}(\boldsymbol{\xi}^\star)$. We can apply Azuma-Hoeffding inequality to prove that, with probability at least $1 - \eta$, it holds $\sum_t |f_t(\boldsymbol{x}_t) - \bar{f}(\boldsymbol{x}_t)| \leq \mathcal{E}_{T,\eta}$ and $\sum_t |f_t(\boldsymbol{\xi}^\star) - \bar{f}(\boldsymbol{\xi}^\star)| \leq \mathcal{E}_{T,\eta}$. Then,

$$\sum_{t=1}^T \bar{f}(\boldsymbol{x}_t) \geq \sum_{t=1}^T f_t(\boldsymbol{x}_t) - \mathcal{E}_{T,\eta}$$

$$\geq \frac{\rho}{1+\rho} \sum_{t=1}^T f_t(\boldsymbol{\xi}^\star) - \left(1 + \frac{2}{\tilde{\rho}}\right) \mathcal{E}_{T,\eta}^{\text{P}} - \frac{1}{\tilde{\rho}} \mathcal{E}_T^{\text{D}} - \mathcal{E}_{T,\eta}$$

$$\geq \frac{\rho}{1+\rho} \sum_{t=1}^T \bar{f}(\boldsymbol{\xi}^\star) - \left(1 + \frac{2}{\tilde{\rho}}\right) \mathcal{E}_{T,\eta}^{\text{P}} - \frac{1}{\tilde{\rho}} \mathcal{E}_T^{\text{D}} - 2\mathcal{E}_{T,\eta},$$

proving the statement. $\qquad \square$

## B.5 Proofs omitted from Section 7

**Lemma 7.1.** *If $T_0 = \sqrt{T}$, then Algorithm 3 guarantees that $\hat{\rho} \leq \rho$ with probability at least $1 - \delta$.*

*Proof.* By Azuma-Hoeffding inequality, we have that with probability at least $1 - \delta$, for each $i \in [m]$ it holds $\left| \sum_{t=1}^{T_0} g_{t,i}(\boldsymbol{x}_t) - \bar{g}_i(\boldsymbol{x}_t) \right|$. Hence,

$$-\max_{i \in [m]} \sum_{t=1}^{T_0} g_t(\boldsymbol{x}_t) \leq -\max_{i \in [m]} \sum_{t=1}^{T_0} \bar{g}(\boldsymbol{x}_t) + \mathcal{E}_{T_0,\delta} \leq T_0 \bar{g}(\boldsymbol{\xi}^\circ) + \mathcal{E}_{T_0,\delta} = T_0 \rho + \mathcal{E}_{T_0,\delta},$$

where the second and third inequality follow from the definition of $\boldsymbol{\xi}^\circ$. Then,

$$\hat{\rho} = -\frac{1}{T_0} \left( \max_{i \in [m]} \sum_{t=1}^{T_0} g_{t,i}(\boldsymbol{x}_t) + \mathcal{E}_{T_0,\delta} \right)$$

$$\leq \frac{1}{T_0} \left( T_0 \rho + \mathcal{E}_{T_0,\delta} - \mathcal{E}_{T_0,\delta} \right)$$

$$= \rho.$$

This concludes the proof. $\qquad \square$

**Lemma 7.4.** *By setting $T_0 = \sqrt{T}$, and assuming that Condition 7.2 is satisfied, after $T_0$ rounds of Algorithm 3 we have that $\hat{\rho} \geq \rho/2$ with probability at least $1 - 2\delta$.*

*Proof.* First, notice that with probability $1 - \delta$, the primal regret minimizer has regret bounded by $\mathcal{E}_{T_0,\delta}^{\text{P}}$. Moreover, by the Azuma-Hoeffding inequality, it holds $\left| \sum_{t=1}^{T_0} \boldsymbol{\lambda}_t g_t(\boldsymbol{\xi}^\circ) - \boldsymbol{\lambda}_t \bar{g}(\boldsymbol{\xi}^\circ) \right| \leq \mathcal{E}_{T_0,\delta}$ with probability $1 - \delta$. Consider the case in which both the conditions hold. This happens with probability at least $1 - 2\delta$ by a union bound.

Then,

$$-\max_{i \in [m]} \sum_{t=1}^{T_0} g_t(\boldsymbol{x}_t) \geq -\sum_{t=1}^{T_0} \langle \boldsymbol{\lambda}_t, g_t(\boldsymbol{x}_t) \rangle - \mathcal{E}_{T_0}^{\text{D}}$$

$$\geq -\sum_{t=1}^{T_0} \langle \boldsymbol{\lambda}_t, g_t(\boldsymbol{\xi}^\circ) \rangle - \mathcal{E}_{T_0}^{\text{D}} - 2\mathcal{E}_{T_0,\delta}^{\text{P}}$$

$$\geq -\sum_{t=1}^{T_0} \langle \boldsymbol{\lambda}_t, \bar{g}(\boldsymbol{\xi}^\circ) \rangle - \mathcal{E}_{T_0}^{\text{D}} - 2\mathcal{E}_{T_0,\delta}^{\text{P}} - \mathcal{E}_{T_0,\delta}$$

$$\geq T_0 \rho - \mathcal{E}_{T_0}^{\text{D}} - 2\mathcal{E}_{T_0,\delta}^{\text{P}} - \mathcal{E}_{T_0,\delta}.$$

Hence,

$$
\begin{aligned}
\hat{\rho} &= -\frac{1}{T_0}\left(\max_{i\in[m]}\sum_{t=1}^{T_0} g_{t,i}(\boldsymbol{x}_t) + \mathcal{E}_{T_0,\delta}\right)\\
&\geq \frac{1}{T_0}\left(T_0\rho - \mathcal{E}_{T_0}^{\mathbb{D}} - 2\mathcal{E}_{T_0,\delta}^{\mathbb{P}} - \mathcal{E}_{T_0,\delta} - \mathcal{E}_{T_0,\delta}\right)\\
&\geq \rho/2 + \frac{1}{T_0}\left(T_0\rho/2 - \mathcal{E}_{T_0}^{\mathbb{D}} - 2\mathcal{E}_{T_0,\delta}^{\mathbb{P}} - 2\mathcal{E}_{T_0,\delta}\right)\\
&\geq \rho/2,
\end{aligned}
$$

where the last inequality comes from Condition 7.2. This concludes the proof. $\qquad\square$

## C  Applications

In this section, we provide more details on possible applications of our techniques to settings related to Internet advertising platforms.

### C.1  Budget-management with ROI constraints

Traditional budget-pacing mechanisms (see, *e.g.*, [8, 11]) are based on primal-dual algorithms which are shown to be near optimal in settings that involve only knapsack constraints, but cannot be generalized to deal with other types of long-term constraints. However, there are many real-world scenarios in which guaranteeing other types of long-term constraints is crucial for practical applications. One example is the case of *return on investment* (ROI) constraints [6, 26, 32]. [9] The recent work by Golrezaei et al. [25] presents a threshold-based algorithms for repeated *second-price* auctions under budget and ROI constraints. Our framework allows advertisers to reach a target ROI while keeping budget expenditures under control also in the setting of repeated *first-price* auctions, which is a frequent setting in practice.[10] In particular, given a target ROI $\omega \geq 0$ and the largest among competing bids $\beta_t$, we define the ROI constraints as

$$
g_t(b_t) = \left(\omega - \frac{v_t}{b_t}\right)\mathbb{1}\{b_t \geq \beta_t\} \leq 0.
$$

Then, it is enough to instantiate the framework with the same setup of Section 8, that is, EXP3.P [5] for each of the possible valuations in $\mathcal{V}$, and OMD equipped with negative-entropy regularizer for the dual RM. Therefore, we immediately get that the cumulative violations of the budget and ROI constraints are upper bounded by $\tilde{O}(T^{1/2})$. This holds both in the fully stochastic and in the fully adversarial setting under the assumption of having a strictly feasible solution, which is reasonable since it is enough to have a sufficiently *small* bid in the set of available bids $\mathcal{B}$. We observe that always bidding such a *small* bid is sufficient to satisfy the ROI constraints but will penalize the cumulative rewards obtained by the advertiser.

### C.2  Future research direction: fairness constraints

Consider the setting in which each item appearing at time $t$ is characterized by one or more of $n_{\mathrm{c}}$ categories according to the vector $\boldsymbol{e}_t \in [0,1]^{n_{\mathrm{c}}}$. A bidder may have distributional preferences over such categories, such as ensuring that at least a certain fraction of impressions is allocated to each category. This is the case, for example, of advertisers who need to perform online outreach to a population of users while achieving a distribution over different demographics *close* to that of the real underlying population. For example, Gelauff et al. [24] provide an interesting field study about running advertising campaigns for Participatory Budgeting elections. In Participatory Budgeting elections, community members are asked to vote between various public projects in order to allocate a total budget. The election organizer may use online advertising to try to promote the initiative,

---

[9]This is a frequent advertising objective in large Internet advertising platform. See, *e.g.*, `https://tinyurl.com/c86rezhd` and `https://tinyurl.com/mr49vz8a`.

[10]For example, in 2019 Google announced a shift to first-price auctions for its AdManager exchange. See `https://tinyurl.com/chv5nxys`.

and in doing so the goal is to reach a "demographic mix" comparable to that of the local population. Surprisingly, Gelauff et al. [24] show that advertisers currently have to resort to complex segmentation strategies through subcampaigns in order to achieve that goal.

Two recent works propose to achieve such distributional preferences within budget-pacing mechanisms by embedding them into a concave regularization term in the advertiser's objective [10, 18]. Such frameworks specifically consider the case of repeated second-price auctions, and can directly handle only packing constraints. Encoding distributional preferences via a regularization term in the objective implies that they cannot provide any formal guarantee w.r.t. how *close* the realized distribution of impressions is to the target, despite showing promising performance in practice.

Differently from previous work, our framework can *explicitly* handle distributional constraints within second- and first-price auction frameworks. Let vector $\hat{\boldsymbol{e}} \in [0, 1]^{n_{\mathrm{c}}}$ be such that $\hat{\boldsymbol{e}}_j$ is the fraction of impressions that we want to be allocated to users of category $j$. Then, for each category $j \in [n_{\mathrm{c}}]$, we could enforce the following type of constraints

$$g_{t,j}(b_t) := \hat{\boldsymbol{e}}_j - \boldsymbol{e}_{t,j} \mathbb{1}\{b_t \geq \beta_t\} \leq 0.$$

Assuming the existence of a strictly feasible bidding strategy, our framework guarantees that, for each category $j$,

$$\hat{\boldsymbol{e}}_j - \frac{1}{T} \sum_{t=1}^{T} \boldsymbol{e}_{t,j} \mathbb{1}\{b_t \geq \beta_t\} \leq \tilde{O}(T^{-1/2}),$$

which guarantees that, in the limit, the difference between the average distribution of impressions and the target thresholds is vanishing.

The main question which still needs to be answered in order to apply our framework in the case of fairness constraints is whether we can motivate the existence of a strictly feasible solution. One reasonable requirement is to constrain the target vector $\hat{\boldsymbol{e}}$ to be a point in the full-dimensional simplex with dimension $n_{\mathrm{c}}$. On top of that, the advertiser would need a way to "buy what's necessary" in order to match the distributional constraints. This desideratum could be achieved, for example, by introducing *buyout options* for advertisers, in the spirit of Gallien and Gupta [23] (*i.e.*, when the advertiser needs impression from a certain category, they always have the option of bidding the buyout value to be sure to win the relevant items). Therefore, assuming the population of users is large enough, an advertiser could achieve a strictly feasible solution by bidding according to the fixed strategy mixture recommending to bid the buyout option for each category $j$ with a probability greater than or equal to $\hat{\boldsymbol{e}}_j$.

The model we described is clearly a simplification of real budget-pacing systems. Moreover, the practical implications of introducing buyout options should be further investigated, in order to understand if they constitute a viable solution both for the platform and advertisers. Finally, we leave as interesting future research directions the problem of studying the general setting (with arbitrary sets $\mathcal{V}$ and $\mathcal{B}$), and that of providing an empirical evaluation of the above techniques on real-world data.