# OpenReview forum: "A Unifying Framework for Online Optimization with Long-Term Constraints"
_NeurIPS.cc/2022/Conference — NeurIPS 2022 Accept_

### Official Review · Reviewer_AWdY · 2022-07-06

**Rating:** 6
**Confidence:** 3
**Soundness:** 4 excellent
**Presentation:** 4 excellent
**Contribution:** 3 good

**Summary:**

The paper proposes a no-regret meta-algorithm that jointly has:
- no constraint regret (i.e. sublinear) if the constraints are stochastic,
- no-$\rho/(1+\rho)$-regret if they are adversarial (i.e. no-regret w.r.t. $\frac{\rho}{1+\rho}\mathrm{OPT}$), where $\rho$ is the strict feasibility margin of the problem.

The main contribution is that the proposed algorithm attains suitable rates across the array of settings considered in past works (hence the title, 'unifying framework'). The paper also studies the application of the algorithm to the repeated auction setting, showing it satisfies "guaranteed budget completion" under certain assumptions (due to guarantees of the algorithm under stochastic constraints *and* rewards).

**Questions:**

- Could your algorithm provide a zero constraint violation guarantee if employed with a safety barrier as in [37] section 3.2? [answered in rebuttal, see below]
- Could you delineate the trade-offs between their algorithm and the main competitors (e.g. does one or the other have better constants? is one or the other easier to implement?)

**Limitations:**

- I think authors should prioritize moving more of the related work discussion from Appendix A to the main body, especially given the nature of the work,
- Similarly, the authors should include a conclusion that clearly address any limitations w.r.t. the existing approaches,
- Could Table 1 be exteded to include more related work? even if the comaprator is different, the comparator type could be noted in an additional column.

Minor Comments:
- should it be 'constraint' rather than 'constraint*s*' in line 4?
- equation $\mathrm{LP}_{f,g}$ could work better as one line?
- maybe change ', e.g. RMs in [41]' to 'RMs such as the ones in [41]' on line 214?
- line 256 : 'allow' -> 'allow*s*'
- line 287: 'to capture also' -> 'to also capture'
- line 339: 'incurring in any cost' -> 'incurring any cost'

**Strengths And Weaknesses:**

Strengths:
- The paper is very well written and provides an elegant general algorithm,
- Having a play phase and a recovery phase is something that I personally haven't seen in the related work (but I have only read a handful of papers in this particular sphere, online optimization with long-term constraints), and it seems to be amendable to downstream analysis of induced properties (e.g. showing the recovery phase is not entered implies the "guaranteed budget completion" property under stochastic constraints and rewards).

Weaknesses:
- there is no *clear* rate improvement with respect to prior work
- the algorithmic approaches and analysis are somewhat standard and expected (given the 'unifying' character of the work this is somewhat natural in my opinion)

---

> ### Author Response · Authors · 2022-08-02
> **Response to Reviewer AWdY**
>
> We thank the Reviewer for the interesting observations and positive comments about the paper. We will fix all the minor comments, and move from the Appendix into the main paper a more detailed discussion of related works (the additional page of content for the camera-ready version could be devoted to that).
>
> - **On the lack of a clear improvement.** We remark that our framework does not only unify previous results already known in the literature, but it also allows us to prove a number of new results. In particular:
> 1. We provide the first guarantees in the adversarial case against the natural baseline of the best fixed decision in hindsight that is feasible for the long-term constraints. Previous work studying the adversarial setting studies weaker baselines, such as the best fixed decision in hindsight that is feasible at every round (i.e., *static regret*, see discussion at lines 580-596).
> 2. We provide, for the first time, an analysis of the case in which $\rho$ is not given as a constant, and it can be arbitrarily small in $T$ or even zero (i.e., Slater’s condition is not satisfied). Our regret and constraint violation bounds of the order of $O(T^{3/4})$ are the first to be proved in such a setting. Notice that previous work considers $\rho$ as a constant and hides it in the big-O notation of the upper bounds; however, this implies that the resulting bounds are arbitrarily bad as $\rho$ becomes small.
> 3. We provide a framework that is able to seamlessly deal with non-convex functions, since it can be instantiated with out-of-the-box regret minimizers working with non-convex losses. We remark that, in order to apply our framework in such settings, we need a suitable regret minimizer for handling non-convex losses for the specific primal problem of interest (as an example, one could employ the regret minimizer by Suggala and Netrapalli [2020]).
>
> - **On the safety barrier as in [37] section 3.2.** We thank the Reviewer for the interesting suggestion. In general, in our setting it is impossible to avoid violating the constraints in the long term (while this is possible in the setting of [37] by exploiting the technique of section 3.2). For instance, if we have $\rho=0$ and the regret minimizer violates the constraint in the first round, it is not possible to recover zero constraint violation in the long term. We conjecture that it could be possible to derive a result similar to the one in [37] by introducing additional assumptions. However, extending the algorithm by [37] to our setting is highly non-trivial since, for example, [37] considers only convex settings, while our framework handles arbitrary reward and constraint functions. Another major difference is that, in [37], the authors focus on the static regret (i.e., the baseline has to be feasible at each round), while we consider the stronger baseline of the best fixed strategy which satisfies in expectation the long-term constraints. Understanding how to extend the techniques by [37] to our setting could be an interesting line of research for future works.
>
> - **On the trade-offs between our algorithm and main competitors.**  It is difficult to provide a precise comparison of the constants in the regret and violation bounds between our works and others. The reason is two-fold:
> 1. First, we provide a general framework without making any specific assumption on the application scenario or on the implementation of the regret minimizers. This means that the regret upper bounds for the algorithm and its practical implementation depend on the specific choices of the regret minimizers. We decided to prioritize a fully general (“unifying”) perspective rather than focusing on a single instantiation of the framework. We leave the problem of evaluating it, also from an experimental perspective, in specific application scenarios as an interesting future research direction.
> 2. Second, previous works either consider different baselines (see, e.g., [35, 42, 43]), make more stringent assumptions (see, e.g., [17, 22]), or provide regret bounds which are not explicit with respect to parameters of interest. For instance, the dependence on the feasibility parameter $\rho$ is usually not considered in previous work, while in our work we tuned our analysis so that it can take into account such dependence. A detailed discussion of these differences is provided in Appendix A. This is also the reason why, in the present version of the paper, we only included [46] in Table 1.
>
> In the final version of the paper, we will provide a more detailed discussion to make the above two points clearer.
>
> References
> - Suggala, Arun Sai, and Netrapalli, Praneeth. "Online non-convex learning: Following the perturbed leader is optimal." Algorithmic Learning Theory. PMLR, 2020

---

> > ### Comment · Reviewer_AWdY · 2022-08-08
> > **Thank you!**
> >
> > Thank you for the through replies to my questions. I agree that despite the fact that no prior rate is explicitly "beaten" there are new implications for different scenarios that may be of interest. I have raised my contribution score from 2 to 3. I also appreciate the clear explanation of the challenges of applying a safety barrier technique to your work, and maybe would suggest adding a small comment on this in a future directions discussion at the end (as I would say it would be very appealing to develop an algorithm that is unifying and also can, under particular settings, simultaneously present this desireable behavior). Finally, please do integrate more of the related work discussion + trade-offs using the additional page for camera-ready.

---

### Official Review · Reviewer_SuMQ · 2022-07-07

**Rating:** 6
**Confidence:** 3
**Soundness:** 3 good
**Presentation:** 2 fair
**Contribution:** 3 good

**Summary:**

The paper deals with the problem of online learning with long term constraints, where the latter are either stochastic  or adversarial. The main contribution is a meta-algorithm based on no-regret online learning for the primal and dual problems, which are solving the "Lagrangian game" introduced by Immorlica et al. (for different type of constraints): one online learner tries to maximize the Lagrangian while the other to minimize it. A novel idea is that, when the constraint violation exceeds a threshold, the algorithm takes care only of the constraints, disregarding the objective function.

The authors prove that their proposed algorithm: (i) achieves, in the case of adversarial constraints, an objective at least a constant fraction of the optimal action in hindsight, while having a constraint violation residual sublinear with T in the case of adversarial constraints where (a bound on) the slack from the Slater condition is known (the constant factor depends on this slack) and (ii) in the case of stochastic constraint achieves a sublinear in T regret and constraint residual even if this bound is not known.

The objective and constraint functions can be non-convex  and the action sets can be non-convex or discrete; this achieved by essentially randomizing the action by finding a probability distribution over the action space (and have the constraints and objectives be the expectation over this probability distribution).

**Questions:**

1. It would be nice to have some examples with some concrete implementations of the Regret Minimizers and their related operations (e.g. in an appendix); this would help towards a better understanding of the algorithms.

For example, in the case of concave rewards and convex costs (and convex action set $\mathcal{X}$), does the algorithm run on a probability distribution over $\mathcal{X}$ or the actions themselves?

2. It would also be nice to have some numerical results about the performance and implementation of the algorithm in the repeated auctions setting of Section 7.

3. In Section 7, why is the auction model restricted to discrete valuations and actions?

**Limitations:**

This is a theoretical work, so I can think of no potential negative social impact.

**Strengths And Weaknesses:**

Strengths:
1. This is the first paper, to my knowledge, to provide performance guarantees for adversarial long-term constraints when the benchmark action is allowed to satisfy the long term constraint (instead of satisfying the constraint in every round or windows of rounds). These guarantees are a step towards understanding more about online learning problems with long term constraints (the result of Mannor et al. just states that the adversary can force $\Omega(T)$ regret if the constraint residual is to be sublinear).
2.  The fact that when constraints are stochastic the slack in the constraints needs not to be known is also a nice result and a nice addition to the related literature.

Weaknesses:
1. The presentation is quite high level and difficult to follow and understand how exactly it works and how it can be implemented.
2. The constant factor depends on the slack $\rho$, which can, at least in theory be very small in the adversarial setting (by the adversary choosing, e.g. constraints that are barely satisfied and/or choosing e.g. $\rho$ as a decreasing function of $T$): it is not clear how useful the approximation bound is in this case.

---

> ### Author Response · Authors · 2022-08-02
> **Response to Reviewer SuMQ**
>
> We thank the Reviewer for the positive comments about the paper and the interesting feedback provided.
>
> - **On the dependence of our upper bound on $\rho$ in the adversarial setting.** See first answer to Reviewer DJUD.
>
> - **[Q1]** In order to deploy our algorithm it is enough to employ any standard regret minimizer for the action set $\mathcal{X}$ for the primal problem. The notion of strategy mixtures is needed to carry out the theoretical analysis of the framework, but in practice the algorithm can be instantiated to work on the original decision space $\mathcal{X}$. In particular, having no-regret with respect to the space of strategy mixtures is equivalent to having no-regret with respect to the action space $\mathcal{X}$ since the primal player problem is linear in the space of strategy mixtures.\
> As an example, in the setting with concave rewards and convex costs with full feedback one could employ *Follow-The-Regularized-Leader* (FTRL, see, e.g., [Hazan & Kale, 2008]) as the primal regret minimizer. Then, the operations used in our pseudo-code would be the following:
> 1. *NextElement*: play the action $x_t$ maximizing the regularized sum of the rewards observed in the past.
> 2. *ObserveUtility*: update the cumulative sum of utilities for each action observed up to time $t$.\
> We will add additional examples in the final version of the paper.
>
> - **[Q2]** We decided to prioritize a theoretical study of the problem. We agree that providing a numerical evaluation in the repeated auction setting is an interesting direction, but we think it would probably be more suited for a work focusing specifically on the repeated auction/online advertising settings, while we tried to keep the scope of the present work as general as possible. Moreover, space constraints would make it difficult to devote enough space to an experimental evaluation. We leave this as an interesting future research direction.
>
> - **[Q3]** We decided to present the discretized setting in order to ease exposition, which would have been more involved if we had to instantiate a primal regret minimizer for the case of continuous valuations/bids. Moreover, works addressing the case of continuous valuations/bids usually require many additional assumptions to provide convergence guarantees (see, e.g., the paper by Han et al. [2020]), so we opted for the simplest setting not requiring additional assumptions. In principle, the framework can be applied also to continuous settings, provided that a primal regret minimizer is chosen appropriately.
>
> References
> - Han, Y., Zhou, Z., Flores, A., Ordentlich, E., and Weissman, T. Learning to bid optimally and efficiently in adversarial first-price auctions. arXiv preprint arXiv:2007.04568, 2020
> - E. Hazan and S. Kale. Extracting certainty from uncertainty: Regret bounded by variation in costs. In Proc. of the 21st Conference on Learning Theory, 2008.

---

### Official Review · Reviewer_DJUD · 2022-07-11

**Rating:** 6
**Confidence:** 2
**Soundness:** 3 good
**Presentation:** 3 good
**Contribution:** 3 good

**Summary:**

This paper studies the problem of online optimization with long-term constraints, where the goal is to maximize the reward and minimize the violation simultaneously. The authors provide a best-of-both-worlds (BOBW) algorithm to handle stochastic and adversarial constraints. This algorithm uses a general regret optimizer (RM) oracle, which can handle the different reward functions and feedback models. For the analysis, a new notion of $\rho$, which measures the margin of strict feasibility, is used to give BOBW performance guarantee.

**Questions:**

~I hope the authors can justify my concern in the weakness part.~

Also, are there any lower bound results based on $rho$, saying that $\frac{\rho}{1+\rho}$ is the best we can do and $O(T^{3/4}$ types of results are the best for the stochastic case? Can the authors explain more about this?

**Limitations:**

Yes, the authors adequately addressed the limitations and there is no potential negative societal impact of their work.

**Strengths And Weaknesses:**

Strength
1. The authors propose a general framework that connects several lines of works based on different types of constraints.

2. The algorithm proposed by the authors is very interesting and intuitive. It consists of two phases: the play phase to select aggressive actions to maximize the reward according to a constraint violation threshold and the recovery phase to compensate for the violation by playing safe actions.

3. The authors give a new notion of $\rho$, and based on $\rho$, the authors provide promising results with weaker assumptions for the stochastic constraint case when $\rho$ is constant. The authors also give results when $\rho$ is not constant and can be arbitrarily small, meaning the constraints are quite difficult to satisfy. Under this case, they provide the first result for the stochastic case, though weaker than the constant case.

Weakness
~1. My main concern is about the adversary case. This work gives a $\frac{\rho}{\rho+1}OPT-O(\sqrt{T})$ reward guarantee, but the authors require $\rho \le O(T^{-1/4})$ and could approach 0 (when $T$ is large), which means the reward guarantee is $0-O(\sqrt{T})$ and the result seems meaningless.~

Post-rebuttal: This concern has been satisfactorily addressed.

---

> ### Author Response · Authors · 2022-08-02
> **Response to Reviewer DJUD**
>
> We thank the Reviewer for the positive comments about the paper and the useful feedback.
>
> - **On the dependence of our upper bound on $\rho$ in the adversarial setting.**  The fact that the algorithm provides guarantees which degrade as $\rho$ gets smaller is expected, and we think it is unlikely that one could provide a similar result without this dependence. This is due to the intrinsic complexity of the adversarial setting in presence of long-term constraints. In order to get an intuition as for why this is the case, it is enough to consider the simpler setting with only budget constraints (see, e.g., [18]). In that setting, the state-of-the-art algorithm provides the same dependence on $\rho$ in the competitive ratio [18], which is known to be tight [12]. Then, we argue that our competitive ratio is essentially optimal since a competitive ratio of $\rho/(1+\rho)$ in our setting is equivalent to the $\rho$ competitive ratio in the budget-constrained case (see discussion at lines 368-374). We leave the problem of providing a formal reduction between the two settings as an interesting future development. Moreover, another interesting direction is to investigate whether it is possible to obtain better bounds by weakening the baseline.
>
> - **On the $O(T^{3/4})$ regret upper bound for unknown $\rho$.** To the best of our knowledge, we are providing the *first* regret upper bound for the case of *dynamic* stochastic constraints and arbitrarily unknown values of $\rho$ (rightmost column of Table 1). Previous works assuming Slater’s condition make the implicit assumption that $\rho$ is a ``reasonably big’’ constant, which is usually hidden in big-O notation as a constant term. However, such results are not meaningful if we allow $\rho$ to take arbitrarily small values, as we do (see discussion in paragraph at lines 58-67). For the case of arbitrarily small values of $\rho$, our $O(T^{3/4})$ upper bound is the first to be independent from $\rho$. This is already a significant improvement with respect to the best previously known bounds for such a setting, which were depending on $\rho$! Providing a lower bound for this setting is a non-trivial open problem, and we leave it as an interesting open question. We conjecture that $O(T^{3/4})$ is the best which one could get by exploiting a primal/dual framework based on the notion of a Lagrangian game as we do.
>
> - **On the weakness highlighted by Reviewer DJUD.** We observe that, in the adversarial setting, we only study the case in which $\rho$ is a *constant* greater than or equal to $T^{-1/4}$ (see Table 1 and Condition 4.5 in Theorem 6.1). Therefore, the lower bound on the reward cannot deteriorate to $0-O(\sqrt{T})$.

---

> > ### Comment · Reviewer_DJUD · 2022-08-04
> > **Thanks for the clarification**
> >
> > Thanks for the detailed explanation and clarification. After reading your response, I understand the high-level difficulty of the adversary case and the unknown $\rho$ case (though it will be nicer if the formal lower bound results are provided to further improve the significance of the current work). I also notice condition 4.5 in Theorem 6.1 which solves my concern. I have updated my review and will keep my score unchanged.

---

### Meta-Review · Area_Chair_7Nrp · 2022-08-27

**Recommendation:** Accept
**Confidence:** Certain

**Metareview:**

This paper studies the problem of online optimization with long-term constraints and proposes an interesting black-box type of algorithm that works well for both the stochastic setting and the adversarial setting. One issue that reviewers brought up for the adversarial case is whether competing with only \rho/(1+\rho) fraction of the optimal utility is indeed meaningful or necessary. In the rebuttal the authors mentioned that this is likely optimal in light of its connection to the literature on the budget-constrained case, but only leave the formal reduction between the two as future directions. We believe that the paper would be much more complete if this reduction is properly spelled out, and thus encourage the authors to do so in the final version.

**Award:**

No

---

### Decision · Program_Chairs · 2022-09-14

Accept